# Graphene/silicon heterojunction for reconfigurable phase-relevant activation function in coherent optical neural networks

Chuyu Zhong [1,7], Kun Liao [2,7], Tianxiang Dai [2], Maoliang Wei[1], Hui Ma[1], Jianghong Wu[3,4], Zhibin Zhang [2], Yuting Ye[3,4], Ye Luo[3,4], Zequn Chen[3,4], Jialing Jian[3,4], Chunlei Sun[3,4], Bo Tang[5], Peng Zhang[5], Ruonan Liu[5], Junying Li[1], Jianyi Yang[1], Lan Li [3,4], Kaihui Liu [2], Xiaoyong Hu [2] ✉ & Hongtao Lin [1,6] ✉

Optical neural networks (ONNs) herald a new era in information and communication technologies and have implemented various intelligent applications. In an ONN, the activation function (AF) is a crucial component determining the network performances and on-chip AF devices are still in development. Here, we first demonstrate on-chip reconfigurable AF devices with phase activation fulfilled by dual-functional graphene/silicon (Gra/Si) heterojunctions. With optical modulation and detection in one device, time delays are shorter, energy consumption is lower, reconfigurability is higher and the device footprint is smaller than other on-chip AF strategies. The experimental modulation voltage (power) of our Gra/Si heterojunction achieves as low as 1 V (0.5 mW), superior to many pure silicon counterparts. In the photodetection aspect, a high responsivity of over 200 mA/W is realized. Special nonlinear functions generated are fed into a complex-valued ONN to challenge handwritten letters and image recognition tasks, showing improved accuracy and potential of high-efficient, all-component-integration on-chip ONN. Our results offer new insights for on-chip ONN devices and pave the way to high-performance integrated optoelectronic computing circuits.

Neuromorphic photonics has attracted extensive attention in recent decades[1]. The light propagation in photonic networks[2,3] achieves the operation of matrix computation and has exhibited the promising potential to break the technical bottleneck of electrical networks, considering that optical devices use photons as information carriers and have the advantages of larger bandwidth, higher information capacity, and lower power consumption. With the prosperity of silicon photonics[4–6], integrated ONNs have achieved exciting accomplishments in artificial intelligent applications including symbol recognition[3,7], vowel analysis[8], image classification[9], etc.

In a neural network, the activation function (AF) introduces nonlinearity, enabling the network to perform complicated tasks, and has an important impact on training speed and computational accuracy[10,11]. For on-chip ONNs without AF devices[12,13], the nonlinear

[1]State Key Laboratory of Modern Optical Instrumentation, College of Information Science and Electronic Engineering, Zhejiang University, Hangzhou 310027, China. [2]State Key Laboratory for Mesoscopic Physics, Frontiers Science Center for Nano-optoelectronics, School of Physics, Peking University, 100871 Beijing, China. [3]Key Laboratory of 3D Micro/Nano Fabrication and Characterization of Zhejiang Province, School of Engineering, Westlake University, Hangzhou, Zhejiang 310024, China. [4]Institute of Advanced Technology, Westlake Institute for Advanced Study, Hangzhou, Zhejiang 310024, China. [5]Institute of Microelectronics of the Chinese Academy of Sciences, 100029 Beijing, China. [6]MOE Frontier Science Center for Brain Science & Brain-Machine Integration, Zhejiang University, Hangzhou 310027, China. [7]These authors contributed equally: Chuyu Zhong, Kun Liao. ✉ e-mail: xiaoyonghu@pku.edu.cn; hometown@zju.edu.cn

operation is carried out by external modulators through computer control[8,9]. This scheme benefits from the flexibility of digital AF selection, but several analog-to-digital conversion steps add latency to the network. A growing number of efforts have been made to develop on-chip AFs[11] in all-optical or electro-optic ways, as shown in Fig. 1a. In all-optical type AF devices, phase change materials (PCM)[3,14,15] or graphene[16,17] are adopted to modify the optical power directly by the optical signal itself through the refractive index or absorption modification. The absence of an electric circuit can help moderate the complexity of network design, but the optical power threshold is relatively large (MW/cm$^2$)[16]. Recently, a non-intrusive germanium-silicon structure[18] can achieve all-optical activation and power monitoring simultaneously, but the nonlinear response is unchangeable, lacking flexibility. Electro-optic type devices can produce reconfigurability. Indium tin oxide (ITO)[19,20] film devices were demonstrated with low power consumption, simple design but extra photodetectors were needed to monitor the signal intensity. Another strategy involves integrating a micro ring resonator (MRR) into Mach-Zehnder interferometer (MZI) circuits with phase shift electrodes[7]. An increasingly popular approach is called light-splitting-and-detection AF unit[21–23], which is adopted in recently reported ONN chips[24–26]. In such AF unit, input optical power is monitored by a PD in an optical bypass, and the photocurrent is transferred to the modulation voltage of a modulator to form a feedback circuit, finally tuning the transmitted optical power. Such a strategy offers high reconfigurability but brings higher power consumption and time delay because of the opto-electric conversion. Nowadays, AF devices or units should seek to achieve smaller power thresholds, lower power consumption, shorter delay, smaller footprints, and higher flexibility. To offer new opportunities to optical AF device, two-dimensional material-assisted silicon photonics has exhibited intriguing potentials[27,28]. Specifically, the synergistic combination of graphene with silicon-based photonic structures has proved its ability to deliver massively enhanced device performances, enriched functionalities and broadened operation waveband[29].

In this article, we point out that the phase shift of an AF device is usually neglected, omitting the fact that the ONN has a complex-valued nature, as illustrated in Fig. 1b. In addition, most classical AFs are not symmetrical over positive and negative values, which is incompatible with positive-only intensity values. Therefore, many classical AFs used in real-valued neural networks are no longer applicable to complex-valued ONNs (More discussed in Section IX in Supplementary Information). Current methods of solving this problem includes applying activation separately on real and imaginary values[30,31], applying activation based on intensity[17,32–35] and applying activation based on phase[36,37]. However, most of the methods often does not account for the crucial relationship between the amplitude and phase of the complex value, which can only be addressed by an activation function that operates on both[38]. Here, we propose a phase-relevant AF device using graphene/silicon (Gra/Si) heterojunction integrated in MRR (Fig. 2a), which functions as modulator and photodetector in a single device. The optical modulation is achieved by plasma dispersion effect of the silicon waveguide[39] and doping of the graphene, which modulate both the resonance wavelength and coupling strength of the MRR. The extensively studied light detecting ability of graphene and graphene/silicon junction[40–43] has also been utilized. Experimentally, a modulation voltage (power) of 1 V (0.5 mW) was obtained in our Gra/Si device, lower than many pure silicon devices[44–46]. In the photodetection aspect, the high responsivity of over 200 mA/W is realized at 1.5 V bias. The dual-functional property allows the device to achieve high reconfigurability. The modulator-detector-in-one feature guarantees shorter time delay, lower energy consumption, and higher integration density than other AF units. In the meanwhile, the MRR provides wavelength-sensitive phase tuning to the AF units. With the mentioned advantages, our devices can create activation functions with unique nonlinearity other than conventional ones[22] with phase-tuning information included (see Table S3 in Section V in the Supplementary Information for quantitative comparison among AF devices). A complex-valued ONN considering phase activation is built in a computer and trained with the phase-activated AFs from our devices, as

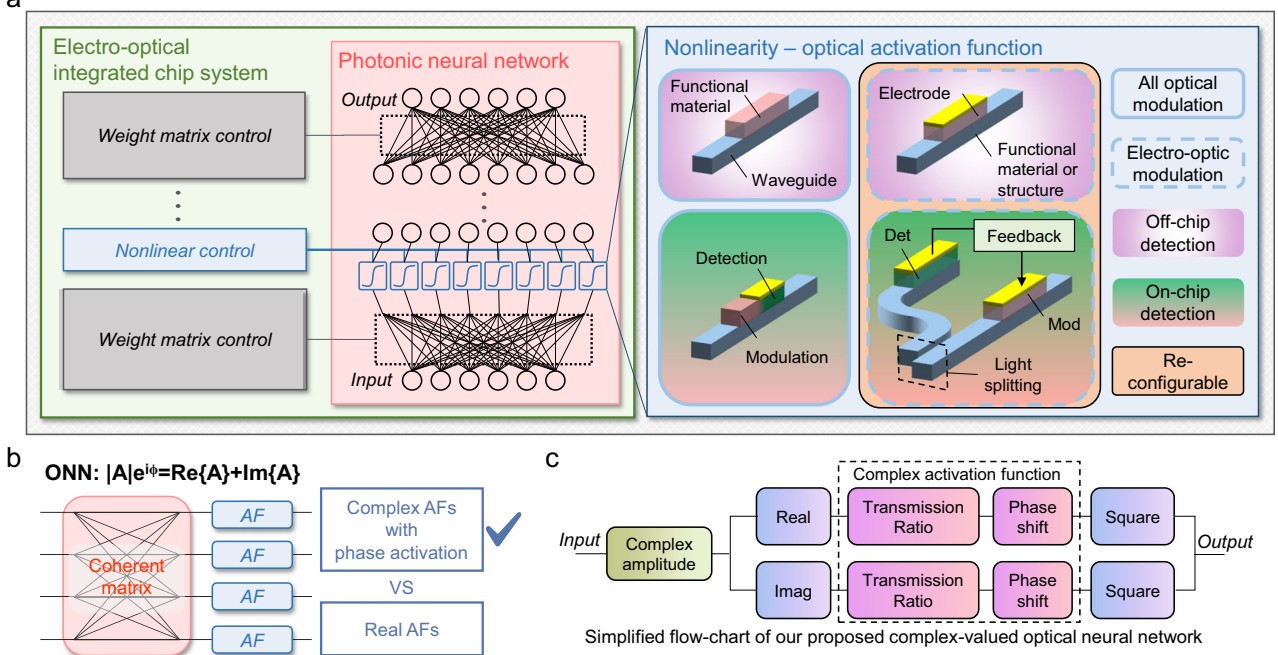

**Fig. 1 | Significance, principle and design of our work. a** General block diagram of photonic neural network integrated activation function devices, where different types of reported on-chip AF devices are compared. **b** Principle of choosing the AF with phase activation based on the fact that the coherent ONN is complex-valued and linear operation matrix is phase-sensitive. **c** Simplified flow-chart of our proposed complex-valued optical neural network. Det detector, Mod modulator, ONN optical neural network, AF activation function, Real real part, Imag imaginary part.

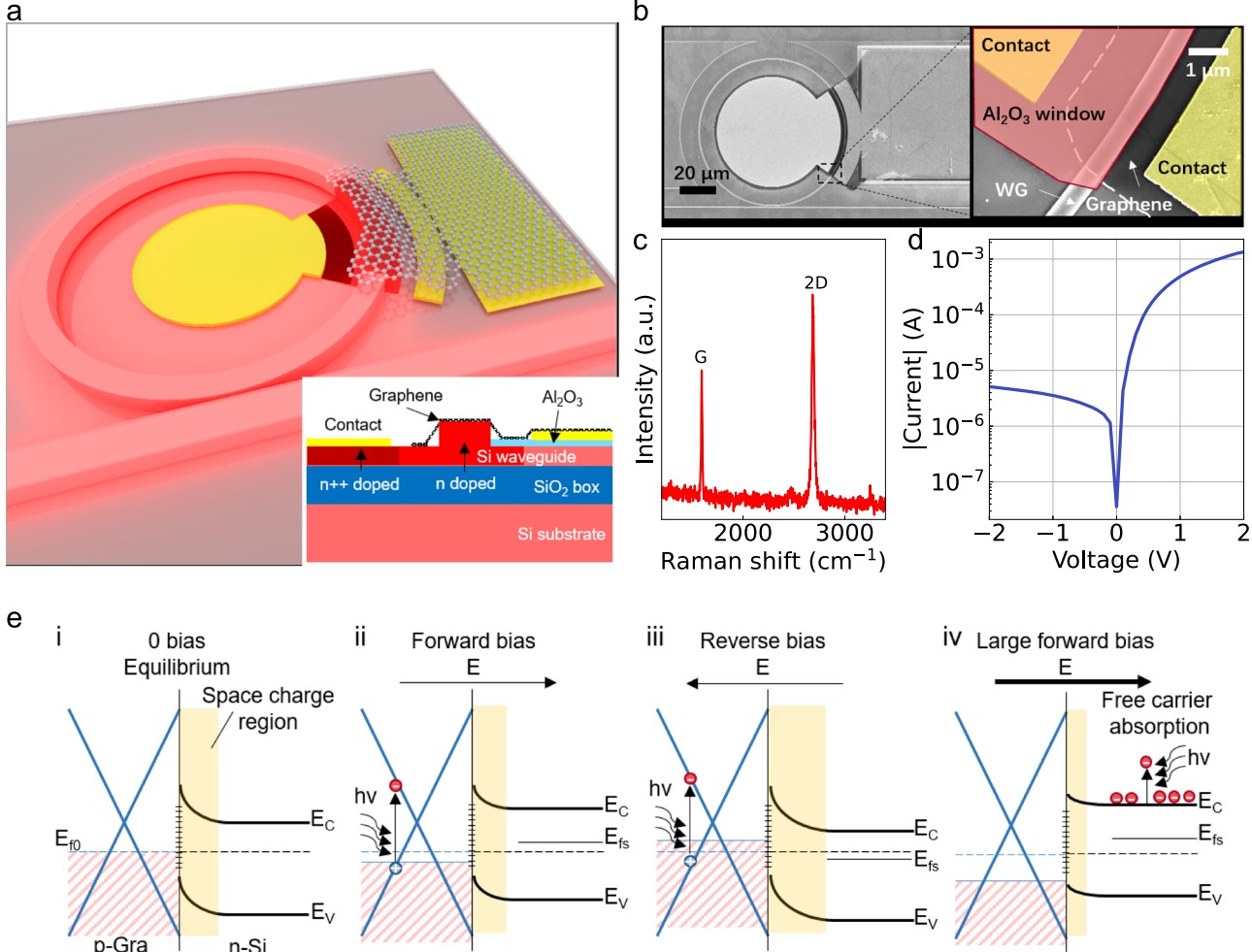

**Fig. 2 | Schematic illustration, properties and operation principle of the graphene/silicon heterojunction. a** Three-dimensional schematic of the graphene/silicon heterojunction device. Inset is the cross-section of the heterojunction structure before photoresist coating. **b** SEM image of the top-view of the heterostructure. **c** Raman spectrum of the graphene. **d** Current-voltage curve of the heterojunction indicating that the device is a heterojunction. **e** Energy-band diagrams of Gra/Si heterostructure. i thermal equilibrium and dark cases at 0 bias. $E_{f0}$, $E_C$ and $E_V$ are the initial Fermi level of the heterojunction, conduction-band bottom and valence-band top of silicon, respectively. ii: band structure under forward bias condition. $E_{fs}$ is Fermi level the silicon. iii: band structure at reverse bias. iv: band structure at large forward bias where the free carrier absorption dominates. The space charge regions are plotted in wheat-colored blocks and the surface states are marked (+) at the Gra/Si interface.

depicted in Fig. 1c. Image classification tasks using MNIST and CIFAR-10 datasets were challenged. Our AFs enable faster convergence speed and higher accuracy. The Gra/Si heterojunction in this work has proposed a positive perspective on future two-dimensional materials photonic networks.

## Results

### Device description and operation principles

The device's structure is illustrated in Fig. 2a and details of the layered device are demonstrated in the inset. Our device was fabricated on a standard silicon photonics platform using a silicon-on-insulator (SOI) substrate by multi-project-wafer (MPW) involved processes (see "Methods"). The photonic structure consists of a ring resonator with a radius of 40 μm. The graphene was transferred monolithically onto the wafer by a standard wet-transfer process. Finally, it formed the graphene/silicon heterojunction with the lightly n-doped waveguide (Fig. 2b). The Raman spectrum in Fig. 2c indicates that the graphene is single-layered and measured current-voltage curve (Fig. 2d) coincides with the electric characteristic of a Schottky diode (more detailed Raman analysis please see Section I in Supplementary Information). In such a Schottky device, carrier engineering can be used to modify the

Fermi level (absorption) of graphene[47] and the refractive index of silicon waveguides[46] (plasma dispersion effect), thereby modulating the optical signal. In the meantime, graphene also functions as a photo-detecting material[48]. The operation principle is explained by the band structure of Gra/Si junction as depicted in Fig. 2e. Under forward bias, the positively charged p-doped graphene has a higher Fermi level and, consequently, is less absorbent. As for the silicon surface, the width of the space charge region is compressed, leading to a larger equivalent doping concentration (smaller refractive index[39]) of the slightly n-doped silicon waveguide. In contrast, graphene is negatively charged under reverse bias, and exhibits increased optical absorption. The space charge region is wider regarding the silicon waveguide, bringing reduced doping concentration (larger refractive index). In the presence of a large forward bias (Fig. 2e.iv), the carrier concentration of the silicon is high, and the free carrier absorption dominates[49]. Such functionalities were demonstrated in ring resonators. With the resonant effect, the modulation power is lower than that of non-resonant structures, and the photodetection is more sensitive due to the light trapping inside. In addition, during the tuning of resonance wavelength, the phase of the output light is also modulated and very sensitive to the position of resonance wavelength (see Section VII in

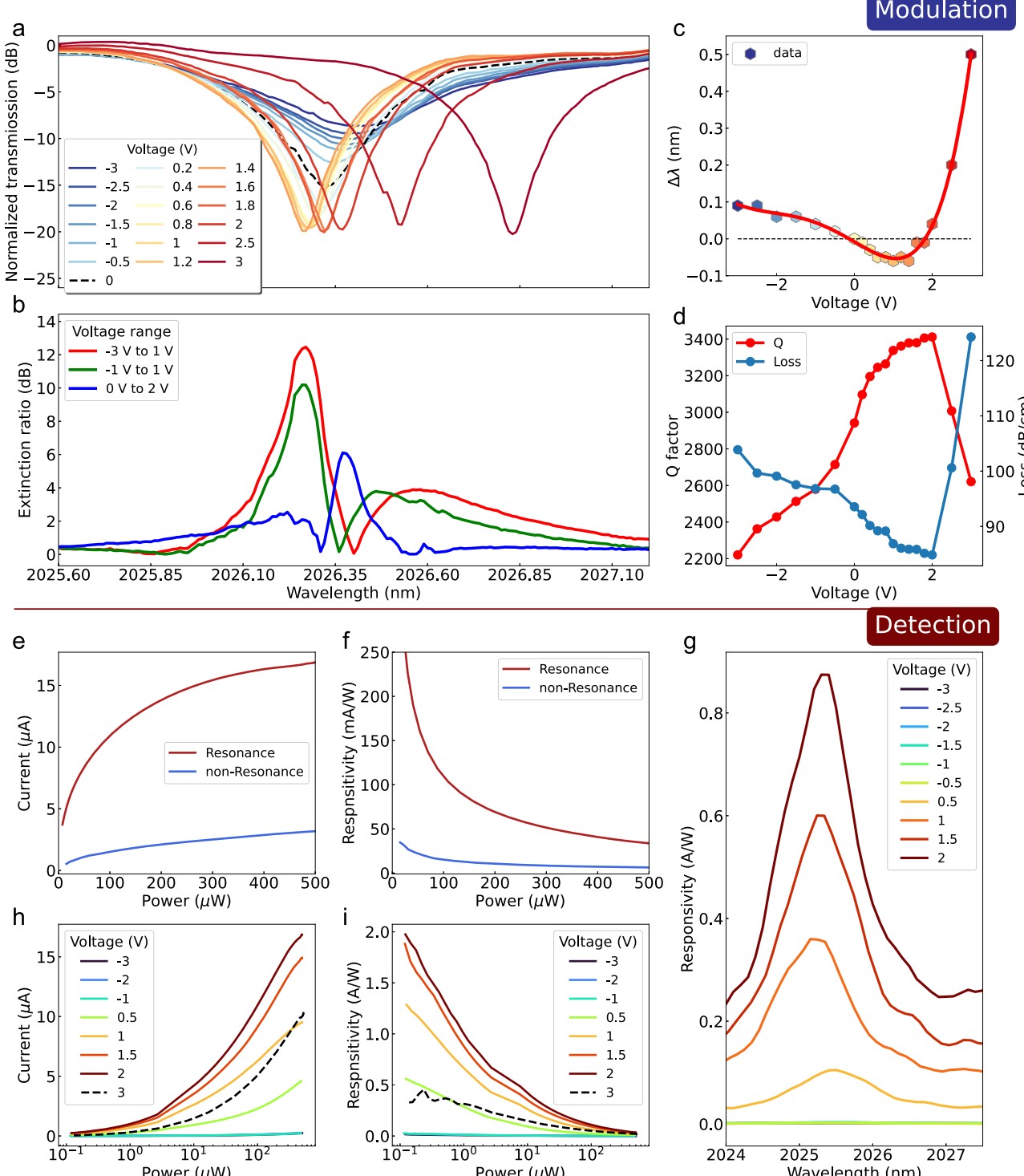

**Fig. 3 | Device performances as both a modulator and a detector. a** Normalized transmission spectra under different voltages. **b** Modulation extinction ratio at different voltage ranges. **c** Wavelength shift at different bias. **d** Q factor and calculated loss (over coupling) at different voltages. **e** Photo-currents and (**f**) responsivity at and not at resonance wavelength. **g** Wavelength-dependent responsivity under different bias. **h** Photo-currents and (**i**) responsivity under different optical power interaction at different bias voltages.

Supplementary Information), exhibiting complex modulation of the optical field.

## Device performances

The modulation performance of the fabricated devices with 50-μm-long graphene (device 1) was characterized, and the results are shown in Fig. 3. The transmission spectra under different voltages (Fig. 3a)

indicate that both the refractive index and the absorption of the active area are tuned by electric driving as discussed. The carrier transfer process differs under different bias conditions; therefore, the effective refractive index($n_{eff}$) and absorption of the active area result in contrasting spectra characterizations. The black dashed curve is the transmission under zero bias. At reverse bias, the resonance wavelength redshifts, and the full width at half maxima of the resonance

peak becomes wider (smaller Q factor as shown in Fig. 3d). A larger refractive index of the silicon waveguide and larger absorption of graphene was calculated in Fig. 3c, coinciding with the results. Under forward bias, the resonance wavelength blueshifts until the bias voltage approaches 1 V, which also agrees with the band structure analysis. In response to increasing voltages over 1 V, the resonance redshifts and shifts faster (Fig. 3c), which could be a result of thermo-optic effects. Hence, our devices can work in carrier injection, carrier depletion, and thermos-optic regions. The modulation depth (extinction ratio) under different voltages below the thermos-optic region is depicted in the lower part of Fig. 2b. Modulation depth exceeding 12 dB can be achieved with a low modulation voltage (power) of about 1 V (0.5 mW), which is smaller than mid-infrared p-n or p-i-n silicon modulators ever reported[50–53]. As for the other two shown modulation operations (−1 V to 1 V and 0 V to 2 V), the largest modulation power is about 2.7 mW, which is also a relatively small value (please see Table S2. in Section IV in Supplementary Information). Then, the detection characterization of our device was performed (Figs. 3e - 2i). As our device is a resonant structure, the photocurrent and responsivity of the resonance wavelength and non-resonance wavelength under a bias of 1.5 V are compared, as shown in Fig. 3e and Fig. 3f. And wavelength-resolved responsivity spectra were measured under different bias voltages (Fig. 3g). Input light in resonance wavelength can produce much larger photocurrent and responsivity. Therefore, our device works as a narrow-band detector. The photocurrent and responsivity at resonance wavelength under different bias voltages and input optical power are illustrated in Figs. 3h and 2i, respectively. Responsivity higher than 200 mA/W can be achieved for input optical power smaller than 100 μW, which exhibits the highest responsivity among the state-of-the-art 2-μm-band graphene-silicon photodetectors, according to the performance comparison in Table S2 in the Supplementary Information. The responsivity for the microwatt-level optical signal can exceed 1 A/W, because the trap states of the graphene-silicon interface prolonged the lifetime of the photoinduced carriers before recombination, leading to the gain which largely improved the responsivity. When optical power increases, the excited electrons contribute to fill the unoccupied states in the graphene to a certain level limited by the photon energy (wavelength). After that, extra incident power (a greater number of photons) will not be absorbed and consequently the photocurrent-power curve become flattened, together with a decreasing responsivity. At 3 V, both photocurrent and responsivity dropped due to a reduced Q factor and increased free carrier absorption (Fig. 3d).

## Generation of activation functions and ONN training

According to the results in Fig. 3, both the output power and photocurrent can be tuned by applying different bias voltages and input optical power. Hence, utilizing the modulation-detection-in-one features of our devices, an on-chip photonic nonlinear activation function with phase tuning for an optical neural network with an ultralow optical power threshold is proposed and validated. The proposed integrated neural network chip system is demonstrated in Fig. 4a. The nonlinearity can be achieved by introducing a photocurrent measurement of the voltage feedback mechanism. An integrated circuit (IC) that can apply bias voltage $V_{in}$ and measure photocurrent $I_p$ can be designed and integrated with the photonic devices so that the bias voltage can be tuned based on the photocurrent variation. Consequently, a transfer function $V'_{in} = H(I_p, V_{in})$ between bias voltage $V_{in}$ and tuned voltage $V'_{in}$ can be programmed into the IC. An easy-to-be-implemented $H(I_p, V_{in})$ is a photocurrent stabilizing circuit. Activation functions were generated from two devices using current stabilizing $H(I_p, V_{in})$. As depicted in Fig. 4b, photocurrent and transmission of device 1 at the wavelength of 2026.31 nm under different voltages and optical input power were obtained. Photocurrent contours of 1 μA and 2 μA are plotted within the filled contour and mapped to the

transmission surface. As a result, the relation between the transmission and input power can be established, and two AFs were extracted and plotted in scattering points. The same operation was performed for device 2 (with a graphene length of 20 μm) at 2012.71 nm, and results are shown in Fig. 4c with three activation functions using photocurrent contours of 0.2 μA and 0.4 μA (more characterization results can be found in Section VI in Supplementary Information). All the AFs with phase shift are demonstrated in Fig. 4d. The phase shift was extracted from equations in Ref. 54 and detailed phase shift deduction is demonstrated in Section VII in Supplementary Information. The configurability of our devices has been proved by the above results that a single device can generate several activation functions by applying different transfer functions related to different photocurrent constants. Even with the same $H(I_p, V_{in})$, different activation functions can also be obtained by choosing different voltage zones. Last but not least, the activation threshold of input optical power as low as 10 μW was achieved, which is order(s) of magnitude lower than other reported results[16,23,55]. Under the above approach, compared to other types of AF devices, our devices can generate complex activation functions with more reconfigurability, simpler operation, lower power consumption and optical threshold (see Table S3 in Section V in the Supplementary Information).

The validity of our optical activation functions is investigated by two complex-valued neural networks in MNIST dataset and CIFAR-10 dataset, respectively. The network structures are illustrated in Fig. S13 in Supplementary Information. The two networks shown in Fig. S13 are based on LeNet[56] and ResNet-34[57], redesigned to adapt to complex-valued convolution and the size of the corresponding dataset. The max pooling layers and fully connected layers of the original network are replaced by a single global average pooling layer, as those layers are unsuitable for optical neural networks. The network's performance is measured in terms of accuracy on the MNIST dataset and CIFAR-10 dataset. Both datasets consist of ten classes with 6000 images per class. The standard train/test split is class-balanced and contains 50,000 training images and 10,000 test images. To monitor the training process, the training set is further split into 40,000 true training images and 10,000 validation images. Images of the CIFAR-10 dataset are RGB-colored with a size of $32 \times 32$ pixels, and images of the MNIST dataset are grayscale with a size of $28 \times 28$ value. We duplicate the real values to both real and imaginary parts for input to the network.

Comparison between our generated optical activation functions against other commonly used activation functions is performed by constructing two complex neural networks for the MNIST dataset and CIFAR-10 dataset. This comparison involved three classical activation functions of real-valued neural networks: Tanh, Arctan and Softsign, our designed optical activation functions, with the identity function (no activation) as the baseline. A diagram of the transmission functions of various activation functions is shown in Section IX in Supplementary Information. We consider the phase shift relative to intensity for our activation functions, and assume it to be 0 for classical ones. We choose $\sqrt{1 - g(x)}$ as our used activation function that operates on the complex amplitude in order to avoid vanishing gradients by increasing the average transmission rate, where $g(x)$ is the transmission rate. The square root corresponds to the relationship between complex amplitude and intensity. A spline interpolation is applied to the data points of our measurement to obtain an analytical piecewise function available for back propagation (see Section IX, X in Supplementary Information).

The training and validation results are depicted in Fig. 5. The training loss (defined as cross-entropy loss) and validation accuracy curves of the complex-valued optical neural networks with different activation functions are demonstrated in Fig. 5b, f. (Comparison between more activation functions can be seen in Section XI in Supplementary Information). The solid dot lines are the average results from 5 training sessions. Our optical activation function shows a much

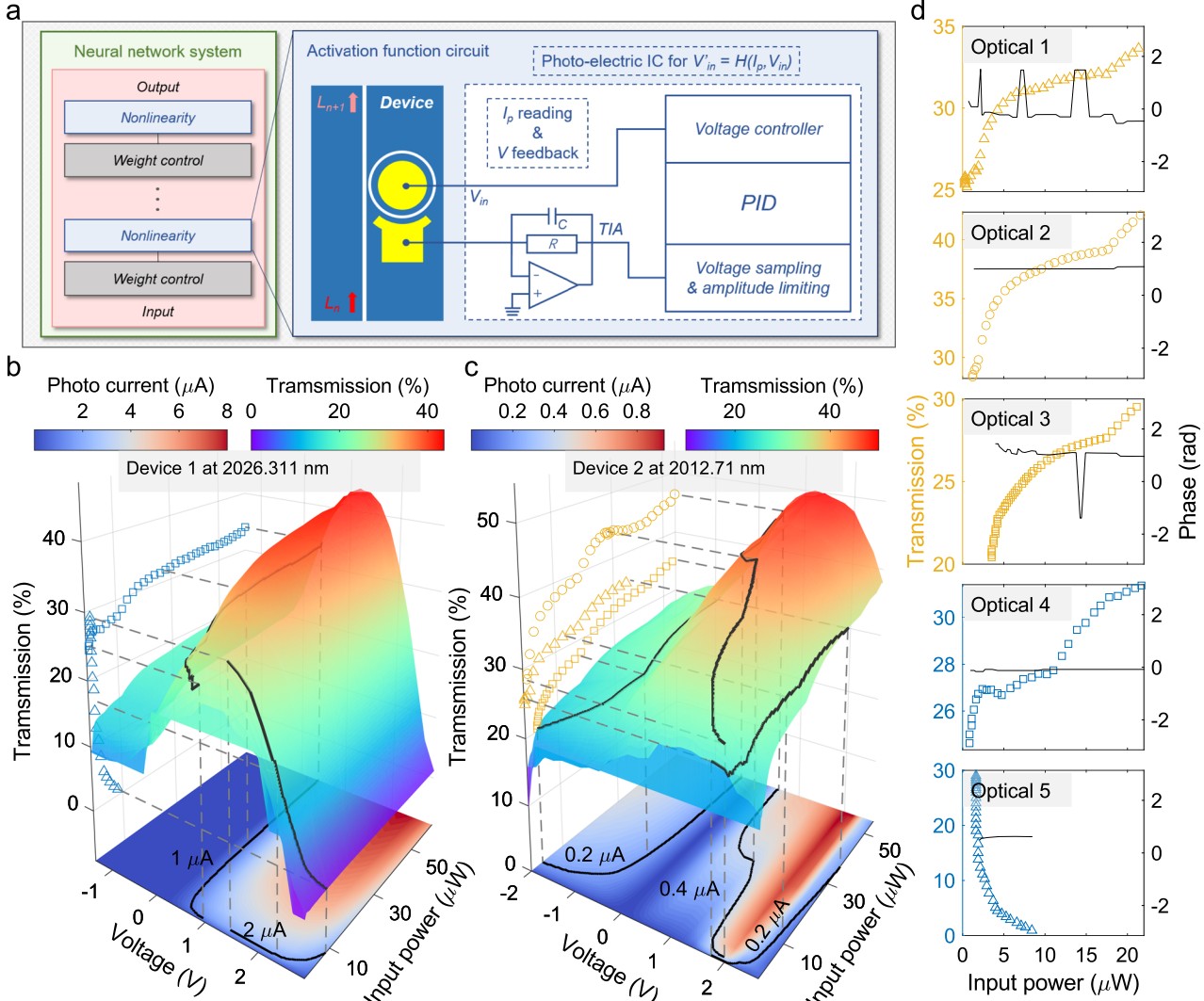

**Fig. 4 | Generation mechanism and results of optical activation fucntions.**
**a** Block diagram of photonic neural network integrated with our activation function devices. **b**, **c** Photocurrent and transmission versus bias voltage and input power of device 1 and device 2, respectively. Black dashed lines in the filled contours of photo current represent the current contours to generate activation functions. Black solid lines in the transmission surface are the corresponding transmission mapped from the photocurrent contours. The scattering plots are data points of the extracted activation functions. **d** Five activation functions with transmission and phase shift information deduced from device 1 and device 2 corresponding to different photocurrent contours.

better loss in the MNIST dataset, indicating a faster convergence speed. The best optical activation function 3 shows a 7% accuracy advantage in both the validation set and test set over the ArcTan (which has the best performance of the classical function in our training) with a loss advantage of 1.5. Moreover, our best optical activation function shows a solid lead over classical functions in the CIFAR-10 dataset, with an 8% accuracy advantage in both the validation set and test set, and converges much faster over the ArcTan, which has the best performance compared with the classical function. It also demonstrates smaller loss values and faster loss reduction versus training rounds. These advantages are due to the transmission rate falling to zero for larger input values for classical functions. A near-zero transmission will result in zero gradient values, prohibiting updating network weights. Besides, it is also possible that the better training results originating from our functions are segmented (Section IX in Supplementary Information), which offers more flexible approximation abilities than smooth functions. The confusion matrices for 10,000 test data set images for different activations are presented in Fig. 5c, g, consistent with the training results. The phase information makes a vital difference in the networks' performance (Section XII in

Supplementary Information). Obviously, our functions can manipulate phase-based intensity, thus taking advantage of complex functions to produce better training results.

A closer analysis of the trained networks is demonstrated in Fig. 5d, h, which shows the visualized output of each block in the neural network, colored based on the intensity values. Our proposed optical activation function shows a much smoother activation map than classical activation functions (see more results in Section XIII in Supplementary Information), with more solid prediction values for the same input compared with the classical activation functions, which proves that the proposed activation function contributes towards stable training of the neural network. A more thorough comparison involving more classical and optical activation functions and the impact of the phase shift can be found in the supplementary materials.

In conclusion, we experimentally demonstrated graphene/silicon heterojunction as modulator and detector in one device, which could operate as a reconfigurable phase-activated optical activation function device to provide a more flexible solution for optical neural networks. The dual-functional devices can be programmed to produce a nonlinear optical response by detecting and modulating the optical signal

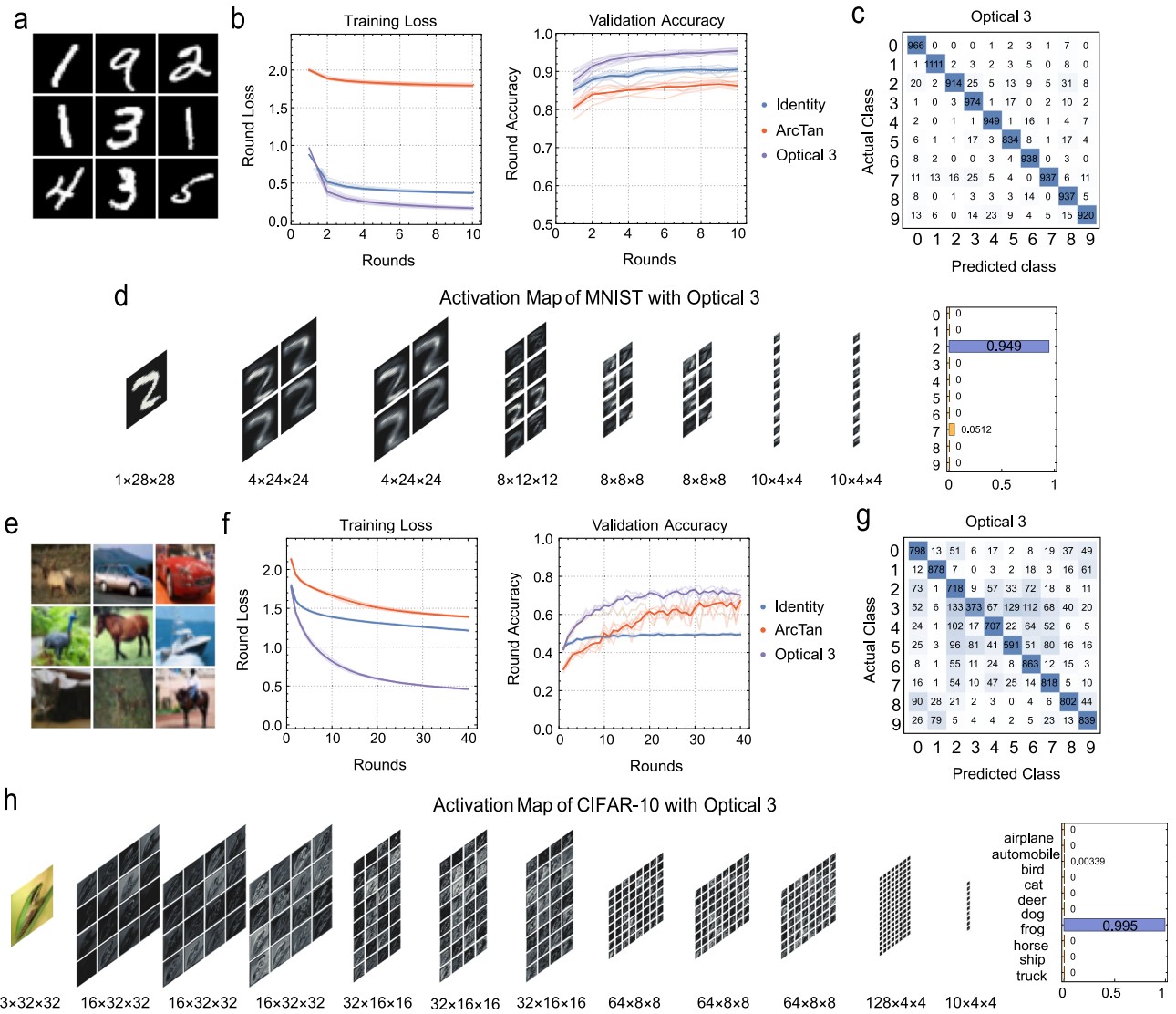

**Fig. 5 | ONN training and results. a** Data examples of MNIST dataset. **b** Training loss and validation accuracy results using different activation functions on the MNIST dataset. **c** Confusion matrix using optical activation function 3 on the MNIST dataset. **d** Visualized activation map trained on the MNIST dataset with the optical activation function 3. **e** Data examples of CIFAR-10 dataset. **f** Training loss and validation accuracy results using different activation functions on the CIFAR-10 dataset. **g** Confusion matrix using optical activation function 3 on the CIFAR-10 dataset. **h** Visualized activation map trained on the CIFAR-10 dataset with the optical activation function 3.

simultaneously. The generated activation functions are more effective and efficient than classical activation functions within the same neural network. The Gra/Si heterojunction on MRR is highly designable and exhibit high reliability (Section VI in the Supplementary Information). Last but not least, as our device can tune the optical intensity, it can also be adopted in the weight matrix part of the optical neural network, which deserves further exploration. We believe this work is promising for future large-scale chip-level optical neural networks.

## Methods
### Device fabrication
The fabrication steps and flowchart are described in detail in Section II in the supplementary information, where structural details of our devices can also befound.

### Device characterization
Please see Section III in the Supplementary Information.

## Data availability
All the data supporting this study are available in the paper and Supplementary Information. Additional data related to this paper are available from the corresponding authors upon request.

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

## Acknowledgements

This work was supported by the National Natural Science Foundation of China (Grant numbers 61975179 received by H.L., 91950204 received by X.H., 92150302 received by D.D., 12104375 received by L.L., 62105287, received by J.L.), the National Key Research and Development Program of China (Grant Number 2019YFB2203002 received by H.L.), the Fundamental Research Funds for the Central Universities (Grant Number 2021QNA5007 received by J.L.). The authors thank ZJU Micro-Nano Fabrication Center at Zhejiang University, Westlake Center for Micro/ Nano Fabrication and Instrumentation, and Service Center for Physical Sciences at Westlake University for the facility support. The authors thank Dr. Min Tao and Prof. Fengli Gao from Jilin University for the feedback circuit diagram of the activation function. We thank Dr. Zhong Chen from Instrumentation and Service Center for Molecular Sciences at Westlake University for the assistance in Raman measurement. We also thank Dr. Chao Zhang from Instrumentation and Service Center for Physical Sciences for the assistance supporting in Hall effect measurement.

## Author contributions

Conceptualization, H.L. and C.Z.; methodology, C.Z and K.L.; software, C.Z. K.L. T.D. and B.T.; validation, C.Z. and K.L.; formal analysis, C.Z.; investigation, C.Z., K.L. T.D., M.W., H.M., Y.Y., J.W., Z.Z., Y.L., Z.C., J.J., C.S., B.T., P.Z., R.L., and J.L.; resources, B.T., P.Z., R.L., Z.Z., K.L. and X.H.; data curation, C.Z.; visualization, C.Z.; supervision, H.L., X.H., K.L., L.L. and J.Y.; All authors contributed to technical discussions and writing the paper.

## Competing interests

The authors declare no competing interests.
