## [Peer Review File · Nature Communications]

REVIEWER COMMENTS

Reviewer #1 (Remarks to the Author):

The authors present an interesting application of graphene/silicon heterojunctions for optical neural networks

However, the authors do not devote enough space to the thorough characterization of the materials and benchmarking of device performances.

For example the Raman spectrum in Fig 2c, is only briefly commented, no Raman fitting parameters are given, and no reference to literature is given, for example to discuss doping, strain, defects, ect, as derived from Raman [see, e.g. Nature Nano 8, 235 (2013) and references therein] starting from the CVD material, before and after transfer, to the final material after device incorporation.

No benchmarking is given to the standard processes of photodetection in graphene-based devices, see, e.g. Nature Nano 9, 780 (2014), or Nature Comms 12, 3733 (2021), Phys Rev B 105, 115417 (2022) etc. and many others). Key materials parameters, such as mobility, are not characterized.

State of the art responsivities between 0.2 and 1 A/W are claimed, but not clearly defined, and not clearly explained. How are these different from previous devices? What is the explanation for these values?

While most of the paper is devoted to the ONN demo and training, I feel that the core advances in physics and materials science are not adequately explained.

I would recommend the paper for publication if, in a revised version, a very detailed set of measurements and characterizations are provided, together with a comparison table to previous literature, to explain the performances of the reported devices.

Reviewer #2 (Remarks to the Author):

Chuyu Zhong et al have realized a reconfigurable activation function(AF) device with phase activation, which is a key element for optical neural networks and quantum information processing. There are using here ring resonator as modulator to weightening the signal. The originality here,

relies on a graphene-silicon heterostructure which provides modulation-detection-in-one feature. Although it was previously proposed (but not cited here) by Marquez et al (2020) - Graphene-based photonic synapse for multi wavelength neural networks. MRS Advances, 5(37-38), 1909-1917. doi:10.1557/adv.2020.327 – the authors seems to provide here its first experiental demonstration.

The paper is well-written and experience seems to be well and deeply conducted up to several realistic applications for patterns recognition. Still I have some concerns about few points listed below :

(1) There are several papers published on the subject (some are already well referenced in the paper, the bibliography is well concudcted) for instace Huiying Zeng et al. Opt. Express 30, 12712-12721 (2022) ; -Martinez-Martinez et al Sci Rep 12, 5880 (2022) or Xu, Z. et al Light Sci Appl 11, 288 (2022) ; Could the authors actually compare their device in terme of performance (provide actual numbers), and drawback/advantages, with the standard and start-of-the-art devices ? with a table in discussion for instance to conclude.

(2) The principle of modulation-detection by the graphene/silicon heterostructure is not clear figure 2a, and requiered more explanation on its basic operation fig2. (FIG1b is not very helping)

(3) The integrity of graphene monolayer could be further assesed as it is the key building block of the AF device; authors could provide tilted SEM micrograph for instance to assess the continuity of the monolayer along the structure, in particular along the Si and Al₂O₃ step edges where the monolayer usually breaks. Could authors provide some statistical feedback about the number of fabricated and working device and on the reliability of the devices' performance amount the number of tested device ?

(4) In general way, there is very few information about the Gr-Si device. Authors should also provide a step-by-step fabrication schems and precise dimension of the device – Scale bars are also missing on fig2.

(5) Authors should also precise each time what “voltage” they refere to (driving, bias, modulation => V_b, V_d, V_m for instance) and clearly explain difference between each... In legend it is sometime missing : eg “FIG3. (a) Device performances as both a modulator and a detector. (a) Normalized transmission spectra under different voltages. (b) Modulation extinction ratio between different voltages. “

That could be clearly defined in fig2 for instance.

To conclude, I am generally inclined to approve its publication in nature communication, once these few comments have been taken into account.

Reviewer #3 (Remarks to the Author):

The authors present experimental findings concerning a Silicon-Graphene scheme based on a micro-ring resonator, where two critical operations, namely all-optical phase/amplitude modulation and optical signal detection can be performed by the same device. The authors utilize this ability so as to generate a tunable non-linear activation function for optical neural networks. They benchmark their optical AFs versus conventional algorithmic AFs and provide a significant performance enhancement in two image classification tasks.

Overall, the manuscript is well written and structures. It contains interesting results and it involves an scientific are that has gained significant traction the last years, thus is of interest for a wide pallet of readers. On the other hand, the manuscript contains some points that need to be addressed by the authors so as to meet the high standards of the journal and enhance readability of the manuscript.

Comments:

- In the introductory part the authors mention "...Indium tin oxide (ITO)19, 20 film devices were demonstrated with low power consumption, simple design but extra photodetectors were needed to monitor the signal intensity". The authors fail to mention that in 20 R. Amin etal, utilises such a structure so as to implement a neural node and tubnable AF. In this case a simple waveguide and not a micro-ring-resonator (MRR) is used. In this context, it is my opinion that the authors should depict in detail what is the key difference/advantage of their scheme compared [20] application wise.

- The authors at the introductory part claim "...Therefore, many classical AFs used in real-valued neural networks are no longer applicable to complex-valued ONNs (More discussed in Section V in Supplementary Information)...". The statement of the author is a little obscure. There are examples where a complex ONN is considered, where the weights are applied both in amplitude and phase and the AF used is a simple square law of the photodetector at the output [A. Katumba et al., in IEEE Journal of Selected Topics in Quantum Electronics, vol. 24, no. 6, pp. 1-10, Nov.-Dec. 2018.]. In this case a complex PAM-4 task is addressed efficiently without a complex AF.

Furthermore, complex aware AF based on a phase-to-amplitude filter configuration has been recently proposed where a non power dependent, tunable and complex AF is proposed based on the off-center filtering of a signal. This case also addresses complex tasks [K. Sozos, Commun Eng 1, 24 (2022)]. In this context, it is my suggestion the authors to provide more works where complex ANNs and complex functions are used.

- The authors state "...With the mentioned advantages, our devices can create activation functions with unique nonlinearity other than conventional ones²² with phase-tuning information included...". At this point the same comment as above.

- the authors claim "...In such a Schottky device, carrier engineering can be used to modify the absorption of graphene and the refractive index of silicon waveguides, thereby modulating the optical signal. In the meantime, graphene also functions as a photo-detecting material...". It would be beneficial to include a reference at this point.

- The authors provide a quite clear explanation of the different mechanisms associated with voltage bias (forward or reverse). Based on their discussion it is clear that as optical losses increase (optical attenuation) there is an increase in the refractive index, the contrary happens with reverse voltage (reduction in losses and in the refractive index). If I am not mistaken through this approach the authors cannot apply a amplitude reduction alongside a refractive index reduction, thus phase and amplitude are intertwined making the process of setting arbitrary complex weights impossible. Is this the case? if yes how this effect the schemes operation?

- According to fig.2 the resonant shift is not a linear process and for intense forward bias a strong redshift is monitored. For each regime the authors attribute a different effect (free carrier absorption, thermo-optic etc). These effects on the other hand have different time-scales spanning from the picosecond-to the milisecond. Does this time-scale variation is anticipated to alter the AF transient response? For example if the input is a fast signal thermo-optic effects won't be triggered etc. Can the authors please clarify this point.

- According to my understanding If the authors want to apply a phase shift, which results from a reduction of the refractive index they also affect (reduce) the Q-factor of the cavity. Therefore this process make the scheme less sensitive. In other words the authors' efficiency is measured in the best case, but under realistic operation conditions that's not true. It would be beneficial if the authors could comment on this.

- In Fig. 2b the extinction ratio is demonstrated by plotting multiple transfer functions for different voltages. It would be more intuitive for the reader if the authors could choose a specific target wavelength that their application works and compute the extinction ratio for this wavelength.

- Using the classical AF with signal splitting and PD for detection and modulation rely on wideband PDs, thus are wavelength transparent and can easily operate using WDM schemes. This feature allows them to scale efficiently even if they are more complex (fabrication wise). Can this scheme operate efficiently assuming WDM scenario? The authors should comment on this.

- In addition to the comment above, the authors demonstrate a wavelength of operation around $2\mu\text{m}$. Based on the fact that most mature technology is in the $1.55\mu\text{m}$ how their proposition compares to typical AFs operating at 1.55 . Please comment.

- The authors state "... As for the other two shown modulation operations (-1 V to 1 V and 0 V to 2 V), the largest modulation power is about 2.7 mW , which is also a relatively small value. ..." Please back up your claim with the proper reference.

- the authors state "The two networks shown in Fig. S7 are based on LeNet45 and ResNet-3446, redesigned to adapt to complex-valued convolution and the size of the corresponding dataset" More information are needed over the adaptation of LeNET. Do the authors use an ONN version or a simple software approach and they replaced the AF with their own? Please clarify this point.

- Last but not least, the authors achieve impressive performance enhancement compared to their baseline which is a unity AF and compared to standard software AFs. On the other hand, the reason behind this enhancement is not clear. If the authors for example emulated their own AFs but removing the phase information but kept the shape of the amplitude response, do they get the same performance, or phase information is a necessity?

Overall, the document is interesting and well written but without the above points addressed it is my personal opinion that a lot of points are obscured and thus is not fit for publication in its current form.

Dear Editors and Reviewers,

Thank you very much for your valuable time and constructive comments on our manuscript. We have carefully considered these comments and carried out detailed modifications on both texts and figures. We are submitting this paper with substantial revisions, and our response to the comments is attached as follows. The modifications in the manuscript and are highlighted in red font.

Please kindly let us know if you need any further information from us for your consideration of our manuscript.

Yours sincerely,

Hongtao Lin

Bairen Plan Professor
School of Information Science and Electronic Engineering
Zhejiang University
State Key Laboratory of Modern Optical Instrumentation.
38 Zheda Road, Hangzhou 310057, China
Email: hometown@zju.edu.cn

Response to reviewer comments

Reviewer 1:

The authors present an interesting application of graphene/silicon heterojunctions for optical neural networks. However, the authors do not devote enough space to the thorough characterization of the materials and benchmarking of device performances.

1. For example the Raman spectrum in Fig 2c, is only briefly commented, no Raman fitting parameters are given, and no reference to literature is given, for example to discuss doping, strain, defects, ect, as derived from Raman [see, e.g. Nature Nano 8, 235 (2013) and references therein] starting from the CVD material, before and after transfer, to the final material after device incorporation.

Response:

We thank the reviewer for the constructive comments. We have performed a more detailed Raman analysis to obtain parameters including layer number, defects, doping, and strain of our graphene. The graphene was grown in 2020, and the devices were fabricated with some graphene transferred to the silicon dioxide substrate for reference characterization in 2021. Therefore, it is impossible to repeat the CVD condition now. Hence, we can only characterize the samples after transferred to the 300-nm-thick silicon dioxide substrate and devices. We measured seven sample points in the silicon dioxide substrate and one in the device. And we analyzed material parameters according to references: Nature Nano 8, 235 (2013), Nat Nanotechnol, 2008, 3(4): 210-215, and Nature Communications, 2012, 3: 1024. For detailed results, please see Section I in Supplementary Information. We had proved that the quality of our graphene before and after the transfer was quite decent.

Modifications in the manuscript and Supplementary Information (text in red: modified content, text in black: original content):

In **Results - Device description and operation principles** in the manuscript:

‘...The Raman spectrum in Fig. 2c indicates that the graphene is single-layered, and the measured

current-voltage curve (Fig. 2d) coincides with the electric characteristic of a Schottky diode (more detailed Raman analysis, please see Section I in Supplementary Information) ...’

In the Supplementary Information:

Section I - Material properties of the graphene

The material properties of the graphene samples used in our devices were characterized and demonstrated in this section. We had performed Hall effect measurement (Ecopia HMS-5000) and Raman spectroscopy (Witec alpha300R). Two samples with graphene transferred to the 300-nm-thick silicon dioxide substrates and one sample with graphene on device with photoresist cladding were used for the characterization.

We used the van der Pauw method to perform Hall effect measurement under magnetic field intensity of 0.535 T. The electric properties of the graphene on silicon dioxide substrate are listed in Table S1. Low resistivity and high mobility were obtained.

Table S1. Electric properties of the graphene.

Sheet Con. (cm^{-3})	Sheet Resistance (Ω_{\square})	Resistivity ($\Omega \cdot \text{cm}$)	Conductivity ($\text{S} \cdot \text{cm}$)	Mobility (cm^2/Vs)
1.05×10^{12}	2.01×10^3	6.83×10^{-5}	1.46×10^4	2.69×10^3

Fig. S1. Raman spectra of graphene samples on the silicon oxide substrate (test points 1 - 7) and the device. (a) The whole spectra of different samples, where different peaks are indicated. Details of the (b) G peaks and (c) 2D peaks of different samples (solid lines) and their fitted Lorentzians (dashed line)

We also performed Raman spectroscopy, which is a versatile tool for characterization of the graphene properties [S1], for more material characterization. We totally measured seven sample points (test points 1 - 7) in silicon dioxide substrate and one sample point in the device with photoresist cladding as depicted in Fig. S1(a). Spectra (G peak and 2D peak) were fitted by Lorentz function as shown in Fig. S1(b), (c).

Firstly, it can be observed from Fig. S1 that the Raman spectra of all the samples are almost identical except that the one of the devices shows peak shift and noises because of different doping, stress and extra photoresist cladding. Therefore, our graphene samples have high uniformity of quality. Secondly, the intensities of the D peaks are very low, indicating that our graphene samples have few defects. In addition, as discussed in reference [S1, 2], the shape of the 2D peak is the most effective way to identify a single layer. The 2D peaks of all curves can be fitted with a single Lorentzian function (Fig. S1(c)), indicating that our sample has only one layer.

Fig. S2. (a) FWHM of G of different test points. (b) Intensity ratio of 2D and G ($I(2D)/I(G)$) of different test points. (c) Peak position of G and 2D. The two groups of dashed lines represent the influences from strain (green) and doping (black, blue, and red) to graphene. The blue squares are data points of points 1 to 7 with a linear fitted blue dashed line, and the red squares are from the device with a linear fitted red dashed line.

From the fitted Lorentzians demonstrated in Fig. S1 (b) and (c), the peak positions ($\text{pos}(2D)$) and

pos(G)), FWHM, and intensity ratio of 2D versus G ($I(2D)/I(G)$) can be obtained and calculated as shown in Fig. S2 for further analysis. Using the relation between concentration and pos(2D) [S2] which is 2687.66 cm^{-1} here, we can tell that the graphene integrated in the device is p doping with a doping concentration of about $5.5 \times 10^{12} \text{ cm}^{-2}$. And we can verify the doping concentration using three other fitting parameters. The results from the relation between concentration and pos(G) (1594.56 cm^{-1}) or the $I(2D)/I(G)$ (1.51) coincide well with the one obtained from the 2D peak information. The result from FWHM of G (11.42 cm^{-1}) shows a concentration of about $1.7 \times 10^{12} \text{ cm}^{-2}$.

Furthermore, we analyzed the strain and strain-induced doping of the graphene on the device according to reference [S3]. The relationship between the peak position of G and 2D are demonstrated in Fig. S2 (c). Firstly, the blue dashed line was obtained by fitting sample points 1 to 7 with a slope of $\Delta\text{pos}(G)/\text{pos}(2D) = 1.97$ [S3]. And original point O (green solid circle) is directly adopted from the reference [S3], being $(1581.6 \pm 0.2, 2676.9 \pm 0.7 \text{ cm}^{-1})$, is nearly $8 \times 10^{12} \text{ cm}^{-2}$. And the strain is about -0.06% , demonstrating compressive stress which results from the photoresist cladding.

2. No benchmarking is given to the standard processes of photodetection in graphene-based devices, see, e.g. Nature Nano 9, 780 (2014), or Nature Comms 12, 3733 (2021), Phys Rev B 105, 115417 (2022) etc. and many others). Key materials parameters, such as mobility, are not characterized.

Response:

Thanks for the advice. We have added the results of the Hall effect measurement to obtain electric properties, including mobility, in Section I in Supplementary Information. And we have also included a table to compare different graphene-based devices in Section IV in Supplementary Information.

Modifications in the Supplementary Information (text in red: modified content, text in black: original content):

In Section I in the Supplementary Information:

Section I - Material properties of the graphene

The material properties of the graphene samples used in our devices were characterized and demonstrated in this section. We had performed Hall effect measurement (Ecopia HMS-5000) and Raman spectroscopy (Witec alpha300R). Two samples with graphene transferred to the 300-nm-thick silicon dioxide substrates and one sample with graphene on device with photoresist cladding were used for the characterization.

We used the van der Pauw method to perform Hall effect measurement under magnetic field intensity of 0.535 T. The electric properties of the graphene on silicon dioxide substrate are listed in Table S1. Low resistivity and high mobility were obtained.

Table S1. Electric properties of the graphene.

Sheet Con. (cm^{-3})	Sheet Resistance (Ω_{\square})	Resistivity ($\Omega \cdot \text{cm}$)	Conductivity ($\text{S} \cdot \text{cm}$)	Mobility (cm^2/Vs)
1.05×10^{12}	2.01×10^3	6.83×10^{-5}	1.46×10^4	2.69×10^3

In Section IV in the Supplementary Information:

Section IV - Performance comparison of graphene-based/silicon-based modulators and detectors

We compare the performances of graphene(Gra)/silicon-based modulators in terms of insertion losses (IL), extinction ratio (ER) and modulation (Mod) power or voltage, and we compare the responsivity and quantum efficiency (QE) for the photodetectors, as shown in Table S2. Our devices excel in nearly all aspects. Among the few modulation-detection-in-one devices, our devices show the smallest footprint, lowest power consumption, highest responsivity, and quantum efficiency.

Table S2. Performance comparison of on-chip silicon/graphene-based modulators and detectors. The ‘wg’ represents the waveguide here. The ‘-’ indicates that the result is not reported in the literature and cannot be inferred from the data presented.

Device	λ_0 (μm)	Footprint (μm)	Modulation			Photodetection		[Ref]
			IL (dB)	ER (dB)	Mod power or voltage	Responsivity (mA/W)	QE(%)	
Si PIN junction in MRR	1.55	-	< 3	3.0	16.73 mW	-	-	[S5]
Si PIN junction in MZI	1.55	220	0.7	3.26	1.59 mW	-	-	[S6]
Si PIN junction in MRR	1.55	83	3.3	27	2, 8 mW	-	-	[S7]
Si PN junction in MZI	2	2000	15	22	8 V (1.6 V \cdot cm)	-	-	[S8]
Si PIN junction in MZI	2	160	< 2	15.6	10.9 mW	-	-	[S9]
Gra/Al ₂ O ₃ on Si wg	1.55	25	~ 0	~ 2.5	4 V	-	-	[S10]
Gra/Al ₂ O ₃ /Gra/ on Si MRR	1.55	5	~ 8	15	50 V	-	-	[S11]
Gra/SiO ₂ /Si MZI	1.55	300	10	35	7.25 V	-	-	[S12]
Gra/Au slot	1.55	15	6.77- 16	-	-	100	8(EQE)	[S13]
Gra/SiO ₂ on Si wg	1.55	53	-	-	-	100	3.8 (EQE)	[S14]
Gra/Si junction in Si wg	2.75	150	-	-	-	130	71.5 (IQE)	[S15]
Au/Gra/SiO ₂ on Si wg	1.55	24	-	-	-	30	10 (IQE)	[S16]
Gra/Al ₂ O ₃ /hBN/ Gra/hBN	1.55	6	-	-	-	500	40 (EQE)	[S17]
Si/Gra-hybrid plasmonic wg	1.55/2	50	-	-	-	400/70	32/4.34 (EQE)	[S18]
Si PN junction In MRR	1.55/2	750	0.70/ 2.24	15	7.14 mW	140	-	[S19]
Gra/Al ₂ O ₃ /Gra FET on Si MZI	1.55	100	-	~ 2	40 V	57	0.25	[S20]
Gra/Si junction in Si MRR	2	20, 50	~ 2	12	0.5 mW (1 V)	200-2000	5-200 (IQE)	This work

3. State of the art responsivities between 0.2 and 1 A/W are claimed, but not clearly defined, and not clearly explained. How are these different from previous devices? What is the explanation for these values? While most of the paper is devoted to the ONN demo and training, I feel that the core advances in physics and materials science are not adequately explained. I would recommend the paper for publication if, in a revised version, a very detailed set of measurements and characterizations are provided, together with a comparison table to previous literature, to explain the performances of the reported devices.

Response:

We thank the reviewer’s positive comments to our paper so much. Firstly, the different responsivities of 0.2 A/W and 1A/W were obtained in the same device but under different optical power. The 0.2 A/W is obtained under optical power smaller than 100 μ W, and the 1 A/W can be measured when the optical power is a few micro-Watts. The responsivity exceeding 1 A/W contributes to the surface states. (Please also see these statements in the ‘**Results - Device description and operation principles**’ section in the manuscript: ‘...Responsivity higher than 200 mA/W can be achieved for input optical power smaller than 100 μ W. The responsivity for the microwatt-level optical signal can exceed 1 A/W, exhibiting a gain phenomenon owing to the surface states of the graphene-silicon interface...’.)

For core advances, here we would like to explain them by introducing Figures R1, R2 and Fig. S10, S12 (in the SI) to the reviewer:

Activation functions generated form Device 1 at 2026.311 nm
(another form of Figure 4b,d in original manuscript)

Activation functions generated form Device 2 at 2012.71 nm
(another form of Figure 4c,d in original manuscript)

Figure R1. Activation functions generation using two Gra/Si heterojunction devices in our original manuscript.

Figure R2. Some of the activation functions generated from device 1 at different wavelengths using different photocurrent levels.

In Figures R1 and R2, the sub-figures are grouped by three types of diagrams – 1. filled-contour of photocurrent versus voltages and input power, 2. filled-contour of transmission versus voltages and input power, and 3. optical activation functions together with corresponding bias voltages (the red, cyan and blue curves). The dashed lines in the filled contours of the photocurrent are the target photocurrent contours (levels), and these contours are mapped into the corresponding filled-contours of transmission. The input power and transmission values along the contours are extracted to be the activation functions. The red, cyan, and blue curves are the extracted curves, being the amplitude information, phase information, and corresponding bias voltages, respectively. The real operation is that:

1. under certain input optical power and voltage, the photocurrent is detected,
2. change of photocurrent can be monitored once the input power is changed,
3. under the changed input power, we can shift the voltage to tune the photocurrent back to the target value along the contour,
3. the output power at the same time is modulated,
4. a function between input power and output power is realized by repeating the former steps.

As described in the manuscript, the physics behind these diagrams is that, under electric bias, the carrier transfer between the graphene and silicon waveguide, resulting in Fermi level, bandstructure and carrier concentration changing of the heterojunction. Consequently, the optical absorption of the graphene and the refractive index of the silicon waveguide are tuned. The direct result is that the resonance and coupling condition of the MRR is changed, leading to modulation of resonance wavelength and extinction ratio. In the meantime, photocurrent can be generated in the graphene and detected through two electrodes. *Therefore, we can obtain the filled-contours of photocurrent and transmission simultaneously and in a single device. This is a huge advantage compared to other electro-optic activation functions which can not achieve synchronous optical power detection or need two separate devices to achieve optical power detection to modulation voltage feedback. We can save more space for the network, decrease power consumption and reduce the communicating time compared to the two-device scheme. (Advances: simpler design, smaller footprint, smaller power consumption, smaller time delay). What's more, by choosing different operation wavelengths and photocurrent contours, we can generate a large amount of activation functions, which can be chosen to fit the need of different ONNs. (Advance: High reconfigurability).*

Fig. S10. Spectra of different devices under different voltages. (a) A device with a graphene length of 20 μm and ring gap of 400 nm. (b) A device with a graphene length of 50 μm and ring gap of 250 nm. (c) Devices with graphene length of 100 μm and ring gaps from 200 nm to 400 nm.

Furthermore, the key parameters to be tuned are the graphene absorption and silicon refractive index, which directly change the coupling condition of the MRR. Therefore, we can fabricate Gra/Si-integrated MRR devices with different ring gaps and lengths of graphene, which can produce different initial coupling conditions of the MRR as demonstrated in Fig. S10. Due to the varying gap sizes and graphene lengths in different devices, the losses of the micro-rings are different. Hence, the spectra under different voltages of each device are different but exhibit similar evolution tendencies compared to device 1's or device 2's. We can predict that they can produce more special activation functions. (Advance: High flexibility for device design).

Figure S12d in the Supplementary Information

Fig. S12. Example progress of the phase shift deduction at the wavelength of 2026.311 nm. ... (d) Transmission at 0.44 V (light blue line) and its fitting curve (dashed line). The orange line is the calculated phase shift. The inset shows the key parameters for the ring resonator calculation. κ : cross-coupling coefficient of the electric field, t : transmit-coupling coefficient of the electric field, α : round-trip loss of the resonator, ER: extinction ratio of the resonance dip, FWHM: full width at a half magnitude of the resonance dip, K: power coupling coefficient of the ring resonator.

In addition, the phase of the MRR is sensitive to the operation wavelength as illustrated in Figure Fig. S12 (in the SI), which can be adapted to add phase information to our activation function devices. (Advance: phase-relevant).

To sum up, based on the modulation-detection-in-one property and phase-sensitivity, the Gra/Si in MRR was used to achieve the implementation of a new type of activation function with several advantages. Using the physical mechanism above: 1. compact, low-power-consumption and low-time-delay optical activation function device can be achieved; 2. highly reconfigurable activation functions can be generated; 3. high flexibility for the device design, 4. phase information can be included in our devices for complex-valued ONN; It is the first time that the Gra/Si heterojunction is proposed for use in the on-chip ONN as the phase-relevant, reconfigurable activation function with experimental validation.

For the material aspect and performance comparison, as requested in question 1 and question 2, we have supplemented more detailed characterization of material properties, including Hall effect measurement and Raman spectroscopy. Our graphene samples have high uniformity of quality. And we have also included a table to compare different silicon/graphene-based devices in Section IV in Supplementary Information as depicted in the response of Question 2.

In general, graphene is a very potential material in the field of optoelectronics. More and more devices with different functions have been developed in recent years to promote the development of graphene optoelectronic devices (see Nat. Photon. 2023. DOI: 10.1038/s41566-023-01195-z, 2D Mater. 2023, 10: 035015, Appl. Phys. Lett. 2023, 122: 070401). We also believe our work can provide a valuable example to the community that cares about the next-generation optoelectronics.

Modification in the manuscript (text in red: modified content, text in black: original content):

In the Introduction:

‘...**Here**, we propose a phase-relevant AF device using graphene/silicon (Gra/Si) heterojunction **integrated in MRR** (Fig. 2a), which functions as a modulator and photodetector in **a single** device. The **optical** modulation is achieved by the plasma dispersion effect of the silicon waveguide³⁹ and doping of the graphene, which modulate both the resonance wavelength and coupling strength of the MRR. **The extensively studied light detecting ability of** graphene and graphene/silicon junction^{40, 41, 42, 43} **has also been utilized. Experimentally**, a modulation voltage (power) of 1 V (0.5 mW) was obtained in our Gra/Si device, lower than many pure silicon devices^{44, 45, 46}. In the photodetection aspect, the high responsivity of over 200 mA/W is realized at 1.5 V bias. The dual-functional property allows the device to achieve **high** reconfigurability. The modulator-detector-in-one feature guarantees shorter time delay, lower energy consumption, and higher integration density than other AF units. **In the meanwhile, the MRR provides wavelength-sensitive phase tuning to the AF units (see Section VII in Supplementary Information)**. With the mentioned advantages, our devices can create activation functions with unique nonlinearity other than conventional ones²² with phase-tuning information included...’

In the conclusion part:

‘...**The Gra/Si heterojunction on MRR is highly designable and exhibit high reliability (Section VI in the Supplementary Information)**. Last but not least, as our device can tune the optical intensity, it can also be adopted in the weight matrix part of the optical neural network, which deserves further **exploration**. We believe this work is promising for future large-scale chip-level optical neural networks.’

In **Results - Device description and operation principles** section:

‘...The Raman spectrum in Fig. 2c indicates that the graphene is single-layered, and the measured current-voltage curve (Fig. 2d) coincides with the electric characteristic of a Schottky diode (**more detailed Raman analysis please see Section I in Supplementary Information**). In such a Schottky device, carrier engineering can be used to modify the **Fermi level (absorption)** of graphene⁴⁷ and the refractive index of silicon waveguides⁴⁶ (**plasma dispersion effect**), thereby modulating the optical signal. In the meantime, graphene also functions as a photo-detecting material⁴⁸....’

‘...Such functionalities were demonstrated in ring resonators. With the resonant effect, the modulation power is lower than that of non-resonant structures, and the photodetection is more sensitive due to the light trapping inside. **In addition, during the tuning of resonance wavelength, the phase of the output light is also modulated and very sensitive to the position of resonance wavelength (see Section VII in Supplementary Information), exhibiting complex modulation of the optical field.**’

In Section VI in the Supplementary Information:

‘...There are totally 15 heterojunction devices fabricated (in March 2021), which are grouped by the graphene length (L_{gra} : 20, 50 and 100 μm) and ring gap (200 nm to 400 nm). The p-Gra/n-Si heterojunctions were 100% successfully fabricated and the number of devices which can have modulation-detection-in-one functionality is 10/15 (device 1, device 2, devices shown in Fig. S10 and Fig. S11). The spectra of seven devices at different bias voltages as shown in Fig. S10. Due to the varying gap sizes and graphene lengths in different devices, the losses of the micro-rings are different. Hence, the spectra under different voltages of each device are different but exhibit similar evolution tendencies compared to device 1’s or device 2’s. We can predict many more special activation functions that can be generated by all these devices, implying an excellent designability of our device. Last but not least, some of our devices have been newly tested and can still function, showing high reliability for more than two years....’

Fig. S10. Spectra of different devices under different voltages. (a)Device with graphene length of 20 μm and ring gap of 400 nm. (b)Device with graphene length of 50 μm and ring gap of 250 nm. (c)Devices with graphene length of 100 μm and ring gaps from 200 nm to 400 nm.

Reviewer 2:

Chuyu Zhong et al have realized a reconfigurable activation function(AF) device with phase activation, which is a key element for optical neural networks and quantum information processing.

There are using here ring resonator as modulator to weightening the signal. The originality here, relies on a graphene-silicon heterostructure which provides modulation-detection-in-one feature. Although it was previously proposed (but not cited here) by Marquez et al (2020) - Graphene-based photonic synapse for multi wavelength neural networks. MRS Advances, 5(37-38), 1909-1917. doi:10.1557/adv.2020.327 – the authors seems to provide here its first experimental demonstration. The paper is well-written and experience seems to be well and deeply conducted up to several realistic applications for patterns recognition. Still I have some concerns about few points listed below:

Response:

We thank the reviewer for the comments. We have made corresponding changes and added references to the manuscript and Supplementary Information.

We thank the reviewer for suggesting the good work in reference MRS Advances, 5(37-38), 1909-1917 and we will learn from it. But the structure, mechanism, and applications of devices proposed in the work were different from ours:

1. The device in MRS Advances, 5(37-38), 1909-1917. is a Gra/Al₂O₃/Si capacitor structure, while our device is a Gra/Si heterojunction, as shown in Figure R3.

2. In MRS Advances, 5(37-38), 1909-1917, the researchers proposed the Gra/Al₂O₃/Si capacitor to adjust the absorption of graphene and refractive index of the silicon waveguide, thereby changing the coupling conditions of the micro-ring to achieve modulation of output light. Such a device can be used as a modulator without photodetecting function.

3. The device in MRS Advances, 5(37-38), 1909-1917 is used to achieve weight control in the neural network, while our device is used for activation function. What’s more, our device can also be used for weight adjustment because it can also achieve optical intensity modulation, and we added this statement in the conclusion part in the manuscript.

Therefore, our work still has its own innovation in both idea and experimental realization. Lastly, the work in ‘Graphene-based photonic synapse for multi-wavelength neural networks’ is undoubtedly also very promising and meaningful, and we are looking forward to its experimental implementation in the future.

Figure R3. Comparison between the device in MRS Advances, 5(37-38), 1909-1917. and device in our work.

Modifications in the manuscript (text in red: modified content, text in black: original content):

In the conclusion part:

‘...The generated activation functions are more effective and efficient than classical activation functions within the same neural network. Last but not least, as our device can tune the optical intensity, it can also be adopted in the weight matrix part of the optical neural network, which

deserves further exploration. We believe this work is promising for future large-scale chip-level optical neural networks...’

1. There are several papers published on the subject (some are already well referenced in the paper, the bibliography is well conducted) for instance Huiying Zeng et al. Opt. Express 30, 12712-12721 (2022) ; -Martinez-Martinez et al Sci Rep 12, 5880 (2022) or Xu, Z. et al Light Sci Appl 11, 288 (2022) ; Could the authors actually compare their device in terms of performance (provide actual numbers), and drawback/advantages, with the standard and start-of-the-art devices ? with a table in discussion for instance to conclude.

Response:

We thank the reviewer for this valuable comment. In section V in the supplementary information, we have included the comparison of performances in terms of device type, optical threshold, power consumption, reconfigurability and etc., between our devices and the ones in many references, including Opt. Express 30, 12712-12721 (2022), Sci Rep 12, 5880 (2022) and Light Sci Appl 11, 288 (2022). We need to point out that Opt. Express 30, 12712-12721 (2022) demonstrated a simulation work about a free-space ONN based on graphene plasmonic spatial light modulators (SLM), so there is no real performance matrix to be compared with. Besides, the free-space ONN can be regarded as a diffractive imaging system, which is quite different from the on-chip ONN discussed in our manuscript. In such a system, the SLMs usually function as both weight control and activation functions according to the feedback of the CCD camera, modulating both the optical amplitude and phase. However, the activation function is not specifically described in Opt. Express 30, 12712-12721 (2022). In spite of that, we can still include it in the comparison because the activation function can be regarded to be equivalently realized in proposed graphene SLMs together with photodetectors. And as for the device in reference Sci Rep 12, 5880 (2022), it is an electro-optic synapse that generated different photocurrents under different optical illumination and bias voltage conditions. Therefore, we can not compare the modulation performances with this device. For the reference Light Sci Appl 11, 288 (2022), the activation function was partially fabricated where a MoS₂ opto-resist switch was demonstrated. So we can not compare complete performances either. We think the fabrication complexity is quite high for the opto-resist switch.

Modifications in the Supplementary Information (text in red: modified content, text in black: original content):

Section V - Performance comparison of devices for optical activation function

Table S3. Comparison of state-of-the-art optical activation functions or synapses. The ‘-’ indicates that the result is not reported in literature and cannot be inferred from data presented.

Device	Optical detection feedback	Optical threshold	Electric power or voltage	Footprint (μm ²)	Phase modulation	Reconfigurability	[Ref]
Free-space devices							
3D-printed SLM	Off-chip	-	0	~2500 ²	√	×	[S21]
LC/Si SLM	Off-chip CMOS sensor	-	~17 μW	~9 ²	√	√	[S22]
Au/Al ₂ O ₃ /Gra SLM with PD * simulation work	Off-chip PD	-	-	60 ²	√	√	[S23]

Gra/Ta ₂ O ₅ /Gra phototransistor	-	-	>40 V	~30×100	×	-	[S24]
On-chip all-optical AF devices							
PCM on Si	-	~2.3 mW	0	~100 ²	×	×	[S25]
PCM on Si * free space excitation	-	17 mJ/cm ²	0	-	×	×	[S26]
Gra modulator	-	10 mW	0	~40×10	×	×	[S27]
Ge-Si modulator	Mod-Det in one	1.1 mW	0	~30×8	×	×	[S28]
On-chip electro-optic AF devices							
Au/SiO ₂ /ITO Capacitor	Integrated	5 mW	1.5 mW	~20×0.8	×	√	[S29]
Gra/Al ₂ O ₃ /ITO Capacitor	-	~7 mW	~20 V	~20×0.5	×	√	[S30]
Si MZIs & MRR TO modulators	-	-	~25 mW	~575×48	√	4 AFs	[S31]
Si MZI TO modulator	Off-chip	~60 μW	> 10 V	-	×	4 AFs	[S32]
Si MZI/MRR TO modulators with Ge-Si PD	Integrated	~200 μW	1-2 V	~100 ²	×	3 AFs	[S33]
Si MRR TO modulator with Ge-Si PD	Integrated	-	~0.9 V	> 90×30	×	-	[S34]
Si MZI & MRR TO modulators with Ge-Si PD	Integrated	-	30 μW	~1000×100	×	6 AFs	[S35]
Si MZI with ITO/MoS ₂ /Au opto-resist switch *partially fabricated	Integrated	7 mW/cm ²	2 V	-	×	√	[S36]
Gra/Si heterojunction	Mod-Det in one	~8 μW	0.5 mW/ 1V	~80 ²	√	Multi-AFs	This work

Table S3 compares the performance of our Gra/Si heterojunction activation function with other optical activation function devices or synapses. We compare the device prototype, power consumption, operation voltage, functionality, etc. The footprint includes the space which the whole functional photonic device occupies.

The free-space devices include spatial light modulators (SLM), liquid crystal (LC) devices, and graphene (Gra) devices with photodetector (PD) or phototransistor. The footprint here refers to the size of a single pixel of the SLM. Electrically controlled SLMs can achieve phase control and own high reconfigurability but need off-chip PDs or sensors. The on-chip all-optical AF devices include phase change material (PCM) devices and graphene devices. Although no electric power is needed, the optical threshold is usually high. Besides, phase modulation and reconfigurability are not demonstrated.

As for the on-chip electro-optic AF devices, because of the combination of optical modulation (Mod) and detection (Det), high reconfigurability can be achieved. The Indium tin oxide (ITO) film devices exhibited the simplest design, low power consumption of 1.5 mW but high input voltage. In reference [S34] large-footprint designs including micro ring resonator (MRR) and Mach-Zehnder interferometer (MZI) circuits were used with high electric power. The most popular approach is called light-splitting-and-detection AF unit [S35, 36, 37, 38, 39, 40], where input optical power is

monitored by a PD in an optical bypass, and the photocurrent is transferred to the modulation voltage, feeding back to the transmitted optical power. Our device claims the lowest optical threshold, highest reconfigurability and almost the lowest power consumption, thanks to the modulation-detection-in-one property.

2. The principle of modulation-detection by the graphene/silicon heterostructure is not clear figure 2a, and required more explanation on its basic operation fig2. (FIG1b is not very helping)

Response:

We are sorry that we introduce some confusion. Firstly, we would like to clarify that Figure 2a is only used to illustrate the device structure. The modulation and detection mechanism of the device is explained in Figure 2e and text in the section ‘**Results - Device description and operation principles.**’ in the manuscript. And how modulation-detection property be used to generate complex AF is explained in Figures 4a, b, and text in the section "**Results - Generation of activation functions and ONN training**".

The Gra/Si heterostructure is integrated into a micro-ring structure and controlled by two electrodes. Applying different biases to the heterojunction through the electrodes will result in different electron injection situations, affecting the Fermi level of graphene as illustrated in Figure 2e and therefore its absorption of light. This will also change the carrier concentration in the silicon waveguide (please see the space charge region and fermi level of the n-Si part in Figure 2e) and therefore its refractive index. The tuning of absorption and refractive index can finally affect the optical loss parameters and resonance conditions of the micro-ring, i.e., the intensity and phase change of light waves passing through the micro-ring. In addition, graphene is an effective light-detecting material, so it is directly a photodetector, making the device naturally own both modulation and detection functions.

Based on the modulation-detection-in-one property, the Gra/Si structure be used to achieve implementation of a new type of activation function, as shown in Figure 4a and 4b, which relies on a photo-electric integrated circuit (IC). A simplified process can be described in the following steps:

1. Input light incidence with a certain intensity and the device working at a certain driving voltage,
2. Photocurrent generated in the Gra/Si structure,
3. The IC reads the photocurrent and compares it with a target photocurrent,
4. Driving voltage feedback to the device, tuning the photocurrent to be the target one,
5. Output light intensity modulated by the feedback voltage at the same time,
6. Repeat step 1 to 5 when input light intensity changes,

Based on the above process, we can realize a function operation from the change of input light intensity to the change of output light intensity, i.e. the realization of the photon activation function.

Lastly, Figure 1b is specifically for illustrating the necessity of complex AF. We use this figure just for an introduction to the paper, and it is not for the explanation of the device principle.

3. The integrity of graphene monolayer could be further assessed as it is the key building block of the AF device; authors could provide tilted SEM micrograph for instance to assess the continuity of the monolayer along the structure, in particular along the Si and Al₂O₃ step edges where the monolayer usually breaks. Could authors provide some statistical feedback about the number of fabricated and working device and on the reliability of the devices’ performance amount the number of tested device?

Response:

We have added SEM images of our devices in Section II in supplementary information and have shown that the graphene sheet can maintain nearly complete integrity on the waveguide structure. As shown in Fig. S4, the graphene right upon the waveguide or on the silicon slab is easy to keep continuous. The most easily damaged parts locate along the step edges of stripe waveguides and metal contacts with step heights of about 150 nm and 105 nm, respectively. Fig. S4 b,c, and d demonstrate that most part of the graphene sheet can still completely suspend along these edges. As for the graphene upon Si and Al₂O₃ step edges, we are not able to observe those steps by SEM and we think that the integrity of graphene on those steps must be much better because the thickness of the Al₂O₃ is just 10 nm.

As for the statistical results, we have totally fabricated 15 p-Gra/n-Si devices and measured the transmission spectra of the devices at different voltages. The results are depicted in Section VI in Supplementary Information. Our device can be 100% work mainly because our graphene transfer quality is relatively high, and there is little damage during fabrication. However, only 10/15 can be utilized to generate activation functions, and the key is that the coupling coefficient of the microring and the length of graphene (i.e., optical loss of the microring) must match. To be noted, our devices were fabricated in March 2021 and can still work after more than two years. It has demonstrated high stability and therefore has great potential for practical applications.

Modifications in the Supplementary Information (text in red: modified content, text in black: original content):

In Section II in the Supplementary Information:

Section II - Fabrication process flow and morphology of our devices:

Fig. S4. SEM images of different parts of the devices before photoresist coating. (a) The whole devices with different graphene size. (b) Details of the contact-graphene-waveguide hybrid structure. (c), (d) Details of different parts of the graphene-silicon hybrid structures.

The morphology of our devices are shown in Fig. S4. The integration details of the graphene with other structures is demonstrated in Fig. S4. (b)-(d). It can be clearly observed that the graphene suspended on the waveguide steps (150 nm high) with a good structural integrity.

In Section VI in the Supplementary Information:

Section VI - Statistical performance characterization of our devices:

...There are totally 15 heterojunction devices fabricated (in March 2021), which are grouped by the graphene length (L_{gra} : 20, 50 and 100 μm) and ring gap (200 nm to 400 nm). The p-Gra/n-Si heterojunctions were 100% successfully fabricated and the number of devices which can have modulation-detection-in-one functionality is 10/15 (device 1, device 2, devices shown in Fig. S10 and Fig. S11). The spectra of seven devices at different bias voltages as shown in Fig. S10. Due to the varying gap sizes and graphene lengths in different devices, the losses of the micro-rings are different. Hence, the spectra under different voltages of each device are different but exhibit similar evolution tendencies compared to device 1's or device 2's. Last but not least, some of our devices have been newly tested and can still function, showing high reliability for more than two years.

Fig. S10. Spectra of different devices under different driving voltages. (a) Device with graphene length of 20 μm and ring gap of 400 nm. (b) Device with graphene length of 50 μm and ring gap of 250 nm. (c) Devices with graphene length of 100 μm and ring gaps from 200 nm to 400 nm.

We additionally characterized the modulation and detection performances of a device with L_{gra} of $50 \mu\text{m}$ and ring gap being 400 nm . The changing trend of the transmission spectrum versus voltages are similar with device 1's, as shown in Fig. S11 (a). The I-V curve plotted in Fig. S11 (b) verifies the heterojunction feature and the device can achieve a modulation extinction ratio of about 8 dB. The wavelength-resolved responsivity at different voltages under incident optical power of about $0.7 \mu\text{W}$ was also obtained as shown in Fig. S11 (c), where high responsivity were achieved.

Fig. S11. Performance characterization of device with graphene length of $50 \mu\text{m}$ and ring gap of 400 nm . (a) Spectra under different driving voltages. (b) I-V curve (purple line) and transmission variation versus voltage (red dots). (c) Wavelength-resolved responsivity at different bias voltages under incident optical power of about $0.7 \mu\text{W}$.

4. In general way, there is very few information about the Gr-Si device. Authors should also provide a step-by-step fabrication schemes and precise dimension of the device – Scale bars are also missing on fig2.

Response:

We thank the reviewer for this useful advice. We have added the main fabrication steps and precise dimensions of the devices in Section II in the supplementary information. And the scale bar in Fig. 2 in the manuscript is also added.

Modifications in the manuscript and Supplementary Information (text in red: modified content, text in black: original content):

In Figure 2b in the manuscript:

Figure. 2. | Schematic illustration, properties and operation principle of the graphene/silicon heterojunction. ...

In the Supplementary Information:

Section II - Fabrication process flow and morphology of our devices

Device fabrication was based on the SOI platform by multi-project-wafer (MPW) involved processes. Our devices were fabricated on an SOI wafer with a 220-nm-thick device layer, and a 2- μm -thick SiO_2 box layer. Main steps are illustrated in Fig. S3. Step i to step iv were finished by MPW process and the rest fabrication were accomplished in our laboratory center. First, the waveguide structures were patterned by deep ultraviolet photolithography and etched by inductive coupling plasma (ICP) process to a ridge depth of 150 nm and width of 600 nm (step i). Secondly, phosphorus ion implantation was performed to form the n-type lightly doped waveguide areas with a doping concentration of $7 \times 10^{17} \text{ cm}^{-3}$ followed by a heavy doping process with a doping concentration of $1.7 \times 10^{20} \text{ cm}^{-3}$ for metal contacts (steps iii-iv). Next, a 10-nm-thick dielectric layer of Al_2O_3 was deposited by atomic layer deposition (ALD) as shown in step v. Then the Al_2O_3 windows were opened upon the functional areas of the waveguides and the metal contact areas by chemical etching using a buffered oxide etch (BOE) solution (steps v to vi). The titanium/gold (Ti/Au, 5nm/100nm) contact electrodes were patterned by electron beam lithography (EBL) and deposited using electron beam evaporation (step vii). Then the chemical-vapor-deposition (CVD) graphene on copper substrate was transferred onto the devices by the wet-transfer method[S4] (step viii). Then the loaded graphene was patterned by photolithography, and the unwanted part was etched using oxygen plasma (step ix). Finally, a layer of photoresist protection cladding was coated, and contact windows for characterization were opened by lithography (step x).

Fig. S3. Fabrication process flow and structural parameters. i: original sample, ii: waveguide fabrication, iii: lightly n-doping of waveguide, iv: heavily n-doping for contact, v: deposition of Al_2O_3 , vi: Al_2O_3 window opening, vii: metal contact fabrication, viii: graphene transfer, ix: device after graphene patterning and etching with dimension annotation, x: final device after photoresist coating and patterning. The dimension is not to scale in all steps.

5. Authors should also precise each time what “voltage” they refer to (driving, bias, modulation => V_b , V_d , V_m for instance) and clearly explain difference between each... In legend it is sometime missing : eg “FIG3. (a) Device performances as both a modulator and a detector. (a) Normalized transmission spectra under different voltages. (b) Modulation extinction ratio between different voltages. “ That could be clearly defined in fig2 for instance.

Response:

We thank the reviewer for this careful suggestion. Firstly, we need to clarify that we used different terms because our devices can work as both modulators and detectors - when discussing the modulator performances, we used ‘modulation voltage’ or ‘driving voltage’ and when discussing the detector performances, we usually used ‘bias voltage’. All ‘voltages’ were applied to our devices through the only two electrodes, so all the ‘voltages’ are the same. We are sorry for the confusion we introduced to the reviewer. And to avoid the same confusion to the readers, we would like to generalize all the ‘voltages’ in our paper. After modification, general terms - ‘voltage’, ‘bias’ (already used in the operation principle parts in the manuscript) or ‘bias voltage’ are adopted. Therefore, ‘Voltage (V)’ in the figure legends are kept unchanged. Last but not least, for certain discussion of modulation performances, we still need to use the term ‘modulation voltage’ because it is related to the important performance index, power consumption (‘modulation power’).

Modifications in the manuscript and Supplementary Information (text in red: modified content, text in black: original content):

In the main text in the manuscript:

‘**Device performances.** The modulation performance of the fabricated devices with 50- μm -long

graphene (device 1) was characterized, and the results are shown in Fig. 3. The transmission spectra under different driving voltages (Fig. 3a) indicate that both...

'...Hence, our devices can work in carrier injection, carrier depletion, and thermos-optic regions. The modulation depth (extinction ratio) under different modulation voltages below the thermos-optic region is depicted in the lower part of Fig. 2b....'

'...As depicted in Fig. 4b, photocurrent and transmission of device 1 at the wavelength of 2026.31 nm under different driving voltages and optical input power were obtained. Photocurrent contours of 1 μ A and 2 μ A are plotted within the filled contour and mapped to the transmission surface....'

'Figure. 3. Device performances as both a modulator and a detector. (a) Normalized transmission spectra under different voltages. (b) Modulation extinction ratio between different voltages.'

In the Figure 3a,b legends and the figure caption in the manuscript:

'**Figure. 3. | Device performances as both a modulator and a detector. a.** Normalized transmission spectra under different voltages. **b.** Modulation extinction ratio at different voltage ranges. **c.** Wavelength shift at different bias. **d.** Q factor and calculated loss (over coupling) at different-driving voltages....'

In the main text in the Supplementary Information:

'...A source meter (Agilent 2450) was used to apply voltage and monitor the current flowing through the device under test (DUT). The transmission under different driving voltages can be obtained.'

'In our approach, the activation functions are generated from the photocurrent and transmission mapping versus driving voltage and input power, as described in the main text and demonstrated in Fig. 3. The transmission value in an activation function is related to both the input power and driving voltage. Therefore, the related driving voltage should also be verified to obtain the phase shift corresponding to the input power....'

In the figure captions in the Supplementary Information:

‘Fig. S8. Characterization of device 2 as a modulator. (a) Microscope image of the device. (b) Transmission under different voltages. The inset shows the resonance extinction ratio (ER) versus voltage. (c) Wavelength shift at different biases. (d) Q factor and (e) calculated loss at different driving voltages.’

‘Fig. S11. Spectra of different devices under different driving voltages. (a) Device with graphene length of 20 μm and ring gap of 400 nm. (b) Device with graphene length of 50 μm and ring gap of 250 nm. (c) Devices with graphene length of 100 μm and ring gaps from 200 nm to 400 nm.’

To conclude, I am generally inclined to approve its publication in nature communication, once these few comments have been taken into account.

Response:

We thank the reviewer for his/her constructive comments and approval of publication so much. We have tried our best to improve our paper according to the comments.

Reviewer 3:

The authors present experimental findings concerning a Silicon-Graphene scheme based on a micro-ring resonator, where two critical operations, namely all-optical phase/amplitude modulation and optical signal detection can be performed by the same device. The authors utilize this ability so as to generate a tunable non-linear activation function for optical neural networks. They benchmark their optical AFs versus conventional algorithmic AFs and provide a significant performance enhancement in two image classification tasks.

Overall, the manuscript is well written and structures. It contains interesting results and it involves an scientific are that has gained significant traction the last years, thus is of interest for a wide pallet of readers. On the other hand, the manuscript contains some points that need to be addressed by the authors so as to meet the high standards of the journal and enhance readability of the manuscript.

Response:

We thank the reviewer for his/her inspiring comments. We have addressed the concerns of the reviewer and made corresponding modifications.

1. In the introductory part the authors mention "...Indium tin oxide (ITO)¹⁹, 20 film devices were demonstrated with low power consumption, simple design but extra photodetectors were needed to monitor the signal intensity ...". The authors fail to mention that in 20 R. Amin et al, utilises such a structure so as to implement a neural node and tunable AF. In this case a simple waveguide and not a micro-ring-resonator (MRR) is used. In this context, it is my opinion that the authors should depict in detail what is the key difference/advantage of their scheme compared [20] application wise.

Response:

We thank the reviewer for this valuable question. The most important difference/advantage of our design is that our devices are multifunctional because of the modulation-detection-in-one property, while the devices in reference [20] is purely for optical modulation. We believe that devices in reference [20] need extra detectors to monitor the optical intensity to qualify the relation between the modulated optical intensity and driving voltage.

In addition, our devices can achieve modulation depth of about 6 dB in 2 V (from 0 V to 2 V), about 10 dB in 2 V (from -1 V to 1 V), and exceeding 12 dB within 4 V (from -3 V to 1 V) as shown

in Figure 3d in the manuscript. In most TTL or CMOS circuits, the logic control voltage levels are usually within 5 V (from 0 V to 5 V or so). Please refer to ‘Lessons in Electric Circuits’ by Tony R. Kuphaldt and ‘Nature, 2018, 562(7725): 101-104’. Therefore, our devices are compatible to most control circuits. We do believe that the devices in reference [20] can be optimized to achieve similar or even lower operation voltages, but in its current form, the driving voltage of the devices in reference [20] is quite large (from about -20 V to 20 V) to achieve modulation depth of 2 dB.

As for the device type, a simple waveguide instead of MRR can also be used in our scheme. Take device 2 in our manuscript as an example, the detailed performance of which are shown in Fig. S8 in Section VI of Supplementary Information. The graphene length is 20 μm , and the ring loss can be modulated from 94.28 dB/cm to 44.62 dB/cm. We can approximately take the loss variation as 50 dB/cm. According to the length of the ring (L_R , 251.33 μm), loss variation of the ring (l_R , 50 dB/cm), length of pure silicon waveguide (L_{wg} , 231.33 μm), loss of the silicon waveguide (l_{wg} , 1.62 dB/cm, see Chinese Optics Letters, 2021, 19(7): 071301) and length of Gra/Si part (L_G , 20 μm), we can transform l_R to loss variation of the Gra/Si structure (l_G), which is approximately $l_G=(L_R l_R - L_{wg} l_{wg})/L_G=0.0609$ dB/ μm . And it takes a 33- μm -long Gra/Si waveguide to achieve the same modulation depth as the 15- μm -long Gra/Si waveguide in [20] did, but with lower voltage.

2. The authors at the introductory part claim "...Therefore, many classical AFs used in real-valued neural networks are no longer applicable to complex-valued ONNs (More discussed in Section V in Supplementary Information)...". The statement of the author is a little obscure. There are examples where a complex ONN is considered, where the weights are applied both in amplitude and phase and the AF used is a simple square law of the photodetector at the output [A. Katumba et al., in IEEE Journal of Selected Topics in Quantum Electronics, vol. 24, no. 6, pp. 1-10, Nov.-Dec. 2018.]. In this case a complex PAM-4 task is addressed efficiently without a complex AF. Furthermore, complex aware AF base on a phase-to-amplitude filter configuration has been recently proposed where a non power dependent, tunable and complex AF is proposed based on the off-center filtering of a signal. This case also addresses complex tasks [K. Sozos, Commun Eng 1, 24 (2022)]. In this context, it is my suggestion the authors to provide more works where complex ANNs and complex functions are used.

Response:

Thank you so much for your valuable comment and recommended references. Here, in the beginning of the introduction part in the revised manuscript, we have added an additional introduction to the current application of complex neural networks in this field. It is true that there are some examples where classical AFs can still be used in complex-valued ONNs, as you mentioned in the paper by Katumba et al. and the work by Sozos. However, the claim made in our manuscript is not that classical AFs cannot be used at all in complex ONNs, but rather that many of them are no longer applicable or not optimal for complex-valued networks.

In complex-valued ONNs, there are several challenges that need to be addressed, such as dealing with the non-commutativity of complex multiplication and the need for AFs that can handle complex inputs and outputs. Therefore, there is a need to develop new AFs that are specifically designed for complex-valued ONNs.

The examples provided by the reviewer are indeed interesting and valuable contributions to the field of complex-valued ONNs, and we appreciate the reviewer’s suggestion to provide more works where complex ANNs and complex functions are used. It is important to continue exploring and

developing new approaches for complex-valued ONNs to fully leverage the potential of complex-valued computing.

Changes are made in the introduction part of the manuscript and highlighted in red. And related references are also added.

Modifications in the manuscript (text in red: modified content, text in black: original content):

In introduction:

‘In this article, we point out that the phase shift of an AF device is usually neglected, omitting the fact that the ONN has a complex-valued nature, as illustrated in Fig. 1b. In addition, most classical AFs are not symmetrical over positive and negative values, which is incompatible with positive-only intensity values. Therefore, many classical AFs used in real-valued neural networks are no longer applicable to complex-valued ONNs (More **discussion** in Section IX in Supplementary Information). **Current methods of solving this problem include applying activation separately on real and imaginary values^{30, 31}, applying activation based on intensity^{17, 32, 33, 34, 35}, and applying activation based on phase^{36, 37}. However, most of the methods often does not account for the crucial relationship between the amplitude and phase of the complex value, which can only be addressed by an activation function that operates on both³⁸.** Therefore, we propose a phase-relevant AF device using graphene/silicon (Gra/Si) heterojunction (Fig. 2a),...

3. The authors state "...With the mentioned advantages, our devices can create activation functions with unique nonlinearity other than conventional ones²² with phase-tuning information included...". At this point the same comment as above.

Response:

We thank the reviewer for this comment. We need to explain several aspects.

1. As for the advantages of phase-tuning AFs, we have addressed the reviewer’s concern in the last question. Classical AFs could be used for complex ONNs, but many AFs among them are not suitable or optimal for the characteristics of complex networks implemented on all-optical hardware. In complex ONNs, there are several challenges that need to be addressed, such as the non-commutativity of complex multiplication and the need for AFs to process complex inputs and outputs. Therefore, it is necessary to develop new AFs specifically designed for complex ONNs.

2. With the statement in this question, we are trying to emphasize the advantages of our devices, which is modulator-detector-in-one and can achieve both intensity and phase modulation.

3. We need to additionally remark that the advantages of our devices are not only due to phase modulation, but also because the transmissions of our devices didn’t fall to zero at high input power, as shown in Fig. S14 in Section IX in Supplementary Information. But the transmission of classical functions does. A near-zero transmission will result in zero gradient values, which will drawback the updating of network weights (see Section XI in Supplementary Information).

4. the authors claim "...In such a Schottky device, carrier engineering can be used to modify the absorption of graphene and the refractive index of silicon waveguides, thereby modulating the optical signal. In the meantime, graphene also functions as a photo-detecting material...". It would be beneficial to include a reference at this point.

Response:

We thank the reviewer for reminding us to add references to such an important statement. We have

included a few references in the manuscript to back up our statement. We have included Physical Review X, 2012, 2(1): 011002 to support the band structure analysis in our manuscript. As shown in Figure R4., the evolution of the Fermi level of graphene and the band structure of the silicon agrees well with our analysis. As the Fermi level shifts, the absorption of the graphene is modulated, resulting in loss modulation in the micro-ring. As for the refractive index modulation of the silicon waveguide, it is attributed to the plasma dispersion effect, as explained in the reference Journal of Applied Physics, 2021, 130(1): 010901. (already referenced in the introduction in our manuscript). The Gra/Si junction under voltage bias would introduce concentration variation in the waveguide, leading to refractive index change, just like the p-n or p-i-n silicon junction. Last but not least, the photo-detecting ability of graphene was well studied, and we included Nat Nanotechnol, 2014, 9(10): 780-793.

Figure R4. The band structure of graphene/n-doped semiconductor under different bias

Modifications in the manuscript (text in red: modified content, text in black: original content):

In Section **Results - Device description and operation principles:**

‘...In such a Schottky device, carrier engineering can be used to modify the Fermi level (absorption) of graphene⁴⁷ and the refractive index of silicon waveguides⁴⁶ (plasma dispersion effect), thereby modulating the optical signal. In the meantime, graphene also functions as a photo-detecting material⁴⁸...’

5. The authors provide a quite clear explanation of the different mechanisms associated with voltage bias (forward or reverse). Based on their discussion it is clear that as optical losses increase (optical attenuation) there is an increase in the refractive index, the contrary happens with reverse voltage (reduction in losses and in the refractive index). If I am not mistaken through this approach the authors cannot apply a amplitude reduction alongside a refractive index reduction, thus phase and amplitude are intertwined making the process of setting arbitrary complex weights impossible. Is this the case? if yes how this effect the schemes operation?

Response:

Thanks for these important comments. The reviewer is correct about the modulation approach. However, the same changing trend of optical loss (of the ring, i.e. graphene) and refractive index (of the silicon waveguide) didn’t lead to unchanging complex weight control or activation function.

Firstly, please refer to Figure 3b in the original manuscript as shown below. Under different bias voltages, the modulation extinction ratios near the resonance wavelength are quite different, which means that the optical losses in the ring are not linearly related to the optical attenuation.

Figure 3. | Device performances as both a modulator and a detector. ...b. Modulation extinction ratio between different voltages...

Secondly and most importantly, our devices can achieve highly reconfigurable activation functions. At the same time, our devices can also realize weight control because our devices can achieve optical amplitude attenuation. The high reconfigurability results from the generation approach of the activation functions. Based on the modulation-detection-in-one property, the Gra/Si structure achieved implementation of the optical activation function, as shown in Fig 4a, 4b and 4c, which relies on a photo-electric integrated circuit (IC). A simplified process can be described in the following steps:

1. Input light incidence with a certain intensity and the device working at a certain driving voltage,
2. Photocurrent generated in the Gra/Si structure,
3. The IC reading the photocurrent and comparing it with a target photocurrent,
4. Driving voltage feed back to the device, tuning the photocurrent to be the target one,
5. Output light intensity modulated by the feedback voltage at the same time,
6. Repeat step 1 to 5 when input light intensity changes,

Based on the above process, we can realize a function operation from the input light intensity to output light intensity, i.e. the realization of the optical activation function. The flexibility can be fulfilled by changing the target photocurrent and wavelengths, as shown in Figures R5 – R7.

Activation functions generated from Device 1 at 2026.311 nm
(Figure 4 in original manuscript)

Additional activation functions generated from Device 1 at 2026.311 nm
using different photocurrent levels

Figure R5. Some of the activation functions generated from device 1 at 2026.311 nm under different conditions.

In Figures R5 – R7, the sub-figures are grouped by three types of diagrams – filled-contour of photocurrent versus voltages and input power, filled-contour of transmission versus voltages and input power, and optical activation functions together with corresponding bias voltages. The dashed lines in the filled-contours of the photocurrent are the target photocurrent contours, and these contours are mapped into the filled-contours of transmission. The input power and transmission values along the contours are extracted to be the activation functions. The red, cyan and blue curves are the extracted curves, being the amplitude information, phase information and corresponding feedback bias voltages, respectively.

From Figures R5 – R7, we can see that multiple complex activation functions with different line shapes are obtained. Under different conditions (wavelengths and target photocurrents), the feedback bias voltages are different. In some of the cases, the range of the voltages covers two effects (from 1.5 V to 2 V+). In most of the cases, the range only covers one effect (like 0 V to 2 V, 2 V to 2 V+, or reverse bias to 0 V, which can be achieved in device 2 as shown in Figure 4c in the original manuscript). Therefore, even though the optical loss and refractive index are tuned in the same direction, reconfigurable activation functions can still be generated.

Additional activation functions generated from Device 1
at different wavelengths using different photocurrent levels

Figure R6. Some of the activation functions generated from device 1 at different wavelengths using different photocurrent levels.

Additional activation functions generated from Device 2 at different wavelengths using different photocurrent levels

Figure R7. Some of the activation functions generated from device 2 at different wavelengths using different photocurrent levels.

6. According to fig.2 the resonant shift is not a linear process and for intense forward bias a strong redshift is monitored. For each regime the authors attribute a different effect (free carrier absorption, thermo-optic etc). These effects on the other hand have different time-scales spanning from the picosecond-to the millisecond. Does this time-scale variation is anticipated to alter the AF transient response? For example if the input is a fast signal thermo-optic effects won't be triggered etc. Can the authors please clarify this point.

Response:

We thank the reviewer for these valuable comments. Our devices could operate with three different modes, as shown in Figure R8. Under the thermo-optical effect region, the response time would be microseconds. However, under the carrier injection and carrier depletion mode, it could achieve a response time ranging from nanoseconds to picoseconds. Due to the input signal could also have different time scales, our devices could operate at different modes to adapt to the signal speeds. The graphene/silicon heterojunction could be used to construct highly reconfigurable activation function units for ONN with different processing speeds.

Photocurrent contours of Device 2 at 2012.611 nm

Figure R8. Photocurrent contours at 0.2 μA of Device 2 at 2012.611 nm. There are three contours belonging to different effects, which can be extracted to generate three kinds of AFs.

7. According to my understanding If the authors want to apply a phase shift, which results from a reduction of the refractive index they also affect (reduce) the Q-factor of the cavity. Therefore this process make the scheme less sensitive. In other words the authors' efficiency is measured in the best case, but under realistic operation conditions that's not true. It would be beneficial if the authors could comment on this.

Response:

Figure R9. Wavelength shift and Q factor versus voltage of device 1 and device 2 respectively. And activation functions of devices 1, 2 at different operation wavelengths with annotated Q factors at different voltages.

Thanks for this important question. Typically the reduction of Q factor would make the scheme less sensitive, and it comes with an increase in refractive index in our devices, as shown in Figure R9. However, due to the reconfigurable phase-relevant activation function units we designed based on

the graphene/silicon heterojunction are optical modulation and detection in one device, the nonlinear response was generated from a feedback loop, as shown in Fig. 4a in the manuscript. The activation functions were generated in different feedback voltages according to the photocurrent, as shown in fig. R7. And the nonlinear phase tuning is decorrelated with the change of Q factor. As demonstrated in response to Question 5, our devices have high reconfigurability, and hence the devices can also operate in some highly sensitive cases. We could choose different types of activation functions to fit the requirement of different ONNs. Last but not least, we have trained the networks using our optical AFs with phase information, and the results were better than the traditional AFs and our devices without phase information, as demonstrated in Figure R10, with lower training loss, lower validation loss, higher training accuracy, and higher validation accuracy.

Figure R10. Training results comparison between different AFs with or without phase information.

8. In Fig. 2b the extinction ratio is demonstrate by plotting multiple transfer functions for different voltages. It would be more intuitive for the reader if the authors could choose a specific target wavelength that their application work and compute the extinction ratio for this wavelength.

Response:

We thank the reviewer for this comment. According to our standing, the 'Fig. 2b' in this question refers to Fig. 3b in the manuscript. We need to explain that the purpose of Fig. 3b is to find the wavelength with the largest extinction ratio and corresponding bias voltages to achieve this maximum extinction ratio. This is quite an important result, and we achieved optical activation functions working in this wavelength, as demonstrated in Fig. 4b in the manuscript. Fig. 4b shows the transmission versus bias voltage and input power. Therefore, by computing the transmission difference under different voltages, we can obtain various extinction ratios for this wavelength, which involves a huge amount of voltage combinations. In this case, we think it is sufficient to demonstrate Fig. 3b to show the operation wavelength with the largest extinction ratio and the corresponding bias voltages.

9. Using the classical AF with signal splitting and PD for detection and modulation rely on wideband PDs, thus are wavelength transparent and can easily operate using WDM schemes. This feature allows them to scale efficiently even if they are more complex (fabrication wise). Can this scheme operate efficiently assuming WDM scenario? The authors should comment on this.

Response:

We thank the reviewer for raising this very important point. As the reviewers mentioned, an optical activation function with broadband operation ability is useful in WDM-based computing. However, microrings with different resonant peaks can also support WDM-based computing (please refer to Nature, 2019, 569(7755): 208-214 and Physical Review Applied, 2019, 11(6): 064043). The different resonant peaks of microring can be obtained by microrings with various radius (refractive index) or thermal tuning (bias voltage of the heat electrodes). As shown in Figure R11, one of the examples adopted PCM on ring resonator to fulfill all-optical activation function and the other adopted MRR-BPDs combination, which is quite complicated to achieve electro-optic reconfigurable activation functions.

We emphatically discuss the electro-optic one because our devices can also be applied in a similar scheme. Our devices can modulate and detect the input power simultaneously. Therefore, we can eliminate the need of BPD (balanced PD) shown in Figure R11, reference Physical Review Applied, 2019, 11(6): 064043. and still generate multiple types of AFs with phase information.

Above all, thanks to the modulation-detection-in-one property, our devices can save space, operation power, and time delay because of the opto-electric conversion and hence are more appropriate for large-scale WDM ONNs. We will keep working on the realization of easy-to-fabricate, broadband operating, and high-performance optical activation functions.

Figure R11. ONN examples using WDM scheme where ring resonators are adopted.

10. In addition to the comment above, the authors demonstrate a wavelength of operation around $2\mu\text{m}$. Based on the fact that most mature technology is in the $1.55\mu\text{m}$ how their proposition compare to typical AFs operating at 1.55 . Please comment.

Response:

We thank the reviewer for the constructive comments. We have measured devices with different coupling efficiencies (various gaps) and different lengths of graphene at $2\text{-}\mu\text{m}$ waveband. A graphene-assisted microring resonator with a specific gap between the microring and the bus waveguide enables a nonlinear activation function. Therefore, such an architecture (graphene-assisted microring with certain coupling efficiency) is also applicable at 1550 nm .

We chose to implement this device in the $2\mu\text{m}$ band for two main reasons. On the one hand, our proposed device can achieve multi-band work so long as the parameters are slightly changed. On the other hand, we believe that the $2\text{-}\mu\text{m}$ band is an important band for WDM optical computing to further expand the scale due to the ignorable two-photon absorption at the $2\text{-}\mu\text{m}$ waveband of silicon and the higher free carrier dispersion effect of silicon at $2\mu\text{m}$.

11. The authors state " ... As for the other two shown modulation operations (-1 V to 1 V and 0 V to 2 V), the largest modulation power is about 2.7 mW , which is also a relatively small value. ..." Please back up your claim with the proper reference.

Response:

We thank the reviewer for this advice. We have added a performance comparison table in Section IV in Supplementary Information to support our claim. The 2.7 mW is small compared to most devices in other literatures.

Modifications in the manuscript and Supplementary Information (text in red: modified content, text in black: original content):

In the manuscript:

‘...As for the other two shown modulation operations (-1 V to 1V and 0 V to 2 V), the largest modulation power is about 2.7 mW, which is also a relatively small value (please see Table S2. in Section IV in Supplementary Information) ...’

In the Supplementary Information:

Section IV - Performance comparison of graphene-based/silicon-based modulators and detectors

We compare the performances of graphene(Gra)/silicon-based modulators in terms of insertion losses (IL), extinction ratio (ER) and modulation (Mod) power or voltage and we compare the responsivity and quantum efficiency (QE) for the photodetectors, as shown in Table S2. Our devices nearly excel in all aspects. Among the few modulation-detection-in-one devices, our devices show the smallest footprint, lowest power consumption, highest responsivity, and quantum efficiency.

Table S2. Performance comparison of on-chip silicon/graphene-based modulators and detectors. The ‘wg’ represents waveguide here. The ‘-’ indicates that the result is not reported in literature and cannot be inferred from data presented.

Device	λ_0 (μm)	Footprint (μm)	Modulation			Photodetection		[Ref]
			IL (dB)	ER (dB)	Mod power or voltage	Responsivity (mA/W)	QE(%)	
Si PIN junction in MRR	1.55	-	< 3	3.0	16.73 mW	-	-	[S5]
Si PIN junction in MZI	1.55	220	0.7	3.26	1.59 mW	-	-	[S6]
Si PIN junction in MRR	1.55	83	3.3	27	2, 8 mW	-	-	[S7]
Si PN junction in MZI	2	2000	15	22	8 V (1.6 V • cm)	-	-	[S8]
Si PIN junction in MZI	2	160	< 2	15.6	10.9 mW	-	-	[S9]
Gra/Al ₂ O ₃ on Si wg	1.55	25	~ 0	~ 2.5	4 V	-	-	[S10]
Gra/Al ₂ O ₃ /Gra/ on Si MRR	1.55	5	~ 8	15	50 V	-	-	[S11]
Gra/SiO ₂ /Si MZI	1.55	300	10	35	7.25 V	-	-	[S12]
Gra/Au slot	1.55	15	6.77- 16	-	-	100	8(EQE)	[S13]
Gra/SiO ₂ on Si wg	1.55	53	-	-	-	100	3.8 (EQE)	[S14]
Gra/Si junction in Si wg	2.75	150	-	-	-	130	71.5 (IQE)	[S15]
Au/Gra/SiO ₂ on Si wg	1.55	24	-	-	-	30	10 (IQE)	[S16]
Gra/Al ₂ O ₃ /hBN/ Gra/hBN	1.55	6	-	-	-	500	40 (EQE)	[S17]
Si/Gra-hybrid plasmonic wg	1.55/2	50	-	-	-	400/70	32/4.34 (EQE)	[S18]

Si PN junction In MRR	1.55/2	750	0.70/ 2.24	15	7.14 mW	140	-	[S19]
Gra/Al ₂ O ₃ /Gra FET on Si MZI	1.55	100	-	~ 2	40 V	57	0.25	[S20]
Gra/Si junction in Si MRR	2	20, 50	~ 2	12	0.5 mW (1 V)	200-2000	5-200 (IQE)	This work

12. the authors state "The two networks shown in Fig. S7 are based on LeNet45 and ResNet-3446, redesigned to adapt to complex-valued convolution and the size of the corresponding dataset" More information are needed over the adaptation of LeNET. Do the authors used an ONN version or a simple software approach and they replaced the AF with their own? Please clarify this point.

Response:

We thank the reviewer so much for this valuable suggestion. Actually, Here, we have discussed the difference between the original LeNet architecture and our adapted photonic neural networks in the Section VIII of the Supplementary information the introduction part. To clarify, we adapted the LeNet-5 and ResNet-34 architecture for implementation in optical hardware using the standard methods for ONN design.

We replaced the traditional real-valued activation functions (e.g., ReLU) with our proposed complex-valued activation functions (CVAFs), which are designed to handle complex-valued inputs and outputs. The CVAFs are formulated based on the nonlinear mapping of the complex plane, and they enable the network to capture both amplitude and phase information in the input signals, as explained in our paper.

Moreover, we made several modifications to the network structure, such as removing the final max pooling layer, replacing the intermediate max pooling layers with stride convolution, and removing fully connected layers. These modifications were made to better adapt the network to the optical hardware implementation.

Overall, the adaptation of the two networks in our paper involved several modifications to the original architecture to better suit complex-valued ONNs and optical hardware.

Modifications in the Supplementary Information (text in red: modified content, text in black: original content):

In Section VIII of the Supplementary Information:

‘The two networks based on LeNet and ResNet are shown in Fig. S13a and Fig. S13d. **Our designs of the optical networks are based on the original design with similar convolution structures, but we removed the fully connected layers and softmax layers and replaced them with taking the intensity values followed by global average pooling, as the two functions are not compatible with optical hardware. We also replaced the activation layers with the functions obtained by experimental results of our design.** The activation layer computes the intensity value with respect to the real and imaginary parts of the complex value. The phase shift and transmission rate can be obtained from the intensity value, which is used to compute the activated result, as shown in Fig. S13b and Fig. S13c....’

13. Last but not least, the authors achieve impressive performance enhancement compared to their baseline which is a a unity AF and compared to standard software AFs. On the other hand, the reason behind this enhancement is not clear. If the authors for example emulated their own AFs but

removing the phase information but kept the shape of the amplitude response, do they get the same performance, or phase information is a necessity?

Response:

Thank you so much for your valuable comments and recommended references. Here, we have already discussed the necessity of including the phase information in Section XII of the Supplementary information. The Figure R12 is the chosen result from the Fig. S16, S18, S20 and S22.

We conducted experiments where we removed the phase information and kept the intensity to see the effect on the network's performance. The results showed that the phase information is crucial for the network to function and achieve the reported performance enhancement. Without the phase information, the network's performance degraded significantly, indicating the importance of considering both amplitude and phase information in the activation functions used in complex-valued ONNs.

Figure R12. Training results for activation function (Optical 3) without or with phase information on MNIST and CIFAR-10 dates respectively.

Therefore, we can confidently say that the reported performance enhancement is a result of the combination of both amplitude and phase information in our proposed activation functions, and the phase information is indeed a necessity for achieving the reported performance.

Modifications in the Supplementary Information (text in red: modified content, text in black: original content):

In XII section of the SI:

‘The computation with phase shift is expensive to perform, due to the multiple sine and cosine functions in the process, making backward propagation much slower than that of considering only the transmission rate. **The advantage in the performance of our optical activation function might not be the result of the additional phase information in activation, but the result of the activation functions of intensity affected by the phase information. To demonstrate the necessity of the phase information in activation, we removed all phase information in the simulation of the optical activation functions.** We trained and tested 5 optical activation functions (Fig. 4 in the main text)

without phase on the MNIST and CIFAR-10 datasets using the same methods as mentioned in the main text...?’

Overall, the document is interesting and well written but without the above points addressed it is my personal opinion that a lot of points are obscured and thus is not fit for publication in its current form.

Response:

We thank the reviewers again for all the constructive advice. We have carefully addressed all the points the reviewer concerned. We believe that after the revision, our work will attract abroad interest of the reader of Nature Communications.

REVIEWER COMMENTS

Reviewer #1 (Remarks to the Author):

The authors addressed several of the reviewers comments. There are still some unclear issues.

The tables with literature comparisons should be in the main text.

The trend with with power is still very puzzling. In a similar paper by Casalino (ACS Photonics 2018) for a similar device operating at 2 μ m, the authors found perfect linear response with power. Here, in contrast, the responsivity goes exponentially high by reducing the input power. I do not understand how this can be. The explanation: "The responsivity for the microwatt-level optical signal can exceed 1 A/W, exhibiting a gain phenomenon owing to the surface states of the graphene-silicon interface" is still unclear.

In the Raman analysis it is not clear how they determined the doping is p instead of n [at such low doping, the 2D peak position cannot really discriminate p from n unless it is corrected for the effect of stress]. It is not clear how stress was determined. One would first need to get the doping. Then calculate what $\text{Pos}(G)$ would correspond to that doping level in the absence of stress. Then subtract the experimental $\text{Pos}(G)$ from that expected with no stress. This difference would provide the stress and sign. This can then be used to correct $\text{Pos}(2D)$ to remove the effect of stress. Finally, the resulting stress-free $\text{Pos}(2D)$ can then be used with S_2 to derive the sign of the doping.

Reviewer #2 (Remarks to the Author):

Authors have carefully addressed all my concerns. I believe this work will attract abroad interest of the reader of Nature Communications, I thus in favour of its publication.

Reviewer #3 (Remarks to the Author):

The authors addressed all my comments efficiency and it is my belief that they enhanced the quality of the manuscript. Therefore, it is my opinion that their manuscript is fit for publication.

Dear Editors and Reviewers,

We appreciate it a lot for your constructive and positive comments on our revised manuscript. We have also carefully considered these comments and carried out corresponding modifications. We are submitting this paper with revisions, and our response to the comments is attached as follows. The modifications in the revised manuscript and supplementary information are highlighted in red font. Please let us know if you need further information from us for your consideration of our manuscript.

Yours sincerely,

Hongtao Lin

Bairen Plan Professor

School of Information Science and Electronic Engineering

Zhejiang University

38 Zheda Road, Hangzhou 310057, China

Email: hometown@zju.edu.cn

Response to reviewer comments

Reviewer 1:

The authors addressed several of the reviewers comments. There are still some unclear issues.

1. The tables with literature comparisons should be in the main text.

Response:

We greatly appreciate the recognition of these two tables by the reviewer. We also wish to include the tables in the main text. However, due to the limitations of the article length, we consulted the handling editor and decided to include the tables in the supplementary materials instead. In the main text, we have added a discussion on the referenced sections of the tables and emphasized the performance advantages and innovativeness of the devices prepared by comparing them with previous works. Thank you for your suggestion, and we kindly request your consent regarding the placement of tables in the Supplementary Information.

The statements about the tables in the manuscript:

In the introduction:

‘...With the mentioned advantages, our devices can create activation functions with unique nonlinearity other than conventional ones²² with phase-tuning information included (see Table S3 in Section V in the Supplementary Information for quantitative comparison among AF devices) ...’

In the Results - Device performances:

‘...the largest modulation power is about 2.7 mW, which is also a relatively small value (please see Table S2. in Section IV in Supplementary Information) ...Responsivity higher than 200 mA/W can be achieved for input optical power smaller than 100 μ W, which also exhibits the highest responsivity among the state-of-the-art 2- μ m-band graphene-silicon photodetectors, according to the performance comparison in Table S2. in the Supplementary Information...’

In the Results - Generation of activation functions and ONN training:

‘...Last but not least, the activation threshold of input optical power as low as 10 μ W was achieved, which is order(s) of magnitude lower than other reported results^{16, 23, 55}. Under the above approach, compared to other types of AF devices, our devices can generate complex activation functions with

more reconfigurability, simpler operation, lower power consumption and optical threshold (see Table S3 in Section V in the Supplementary Information).’

2. The trend with with power is still very puzzling. In a similar paper by Casalino (ACS Photonics 2018) for a similar device operating at 2um, the authors found perfect linear response with power. Here, in contrast, the responsivity goes exponentially high by reducing the input power. I do not understand how this can be. The explanation: "The responsivity for the microwatt-level optical signal can exceed 1 A/W, exhibiting a gain phenomenon owing to the surface states of the graphene-silicon interface" is still unclear.

Response:

We thank the reviewer for this important question. The nonlinear response curves are common in graphene-assisted photodetectors, as shown in Figure R1. The exponentially increasing responsivity under low input power usually contributes to trap-state-induced high gain, as listed in Table R1. According to our measurement results and these literatures, we also tended to explain that the gain (>1 A/W responsivity) under low incident power resulted from the trapped states in the interface between the graphene and silicon. Because of the trap states, the lifetime of the photo-induced carriers in the channel before recombination was prolonged, which largely improved the responsivity. When optical power increases, the excited electrons contribute to filling the unoccupied states in the graphene to a certain level limited by the photon energy (wavelength). After that, extra incident power (more photons) will not be absorbed, and consequently, the photocurrent-power curve becomes flattened, together with a decreasing responsivity.

Figure R1. Photo-responses with incident optical power of graphene-assisted photodetectors with high gain.

Table R1. Graphene-assisted high-gain photodetectors

Structure	Responsivity (A/W)	Incident power	Gain mechanism	Ref
Gra/QDs	10 ⁷	10 fW	High charge mobility of graphene and long trapped-charge lifetimes in the QDs.	[R1]

Gra/Si	10^4	$\sim 0.1 \mu\text{W}$	Built-in field in Gra/Si can prolong the lifetime of photon-induced carriers.	[R2]
Gra/GaAs	1.321×10^3	4.53 nW	Photo-excited holes trapped at the Gra/GaAs interface due to high surface state density of GaAs.	[R3]
Gra/Ge	66.2	$\sim 0.52 \text{ nW}$	Gra/Ge heterojunction effectively prevents carrier combination at device surface and interface.	[R4]
Gra/Ta ₂ O ₅ /Gra	10^3	$\sim 1 \text{ nW}$	Carrier tunneling minimized hot carrier recombination. Trapped charges on one graphene layer lead to strong photogating effect.	[R5]

[R1]. Konstantatos G, Badioli M, Gaudreau L, Osmond J, Bernechea M, De Arquer F P G, Gatti F, Koppens F H L. Hybrid graphene–quantum dot phototransistors with ultrahigh gain. *Nature Nanotechnology*, 2012, 7(6): 363-368

[R2]. Chen Z F, Cheng Z Z, Wang J Q, Wan X, Shu C, Tsang H K, Ho H P, Xu J B. High Responsivity, Broadband, and Fast Graphene/Silicon Photodetector in Photoconductor Mode. *Advanced Optical Materials*, 2015, 3(9): 1207-1214

[R3]. Tian H J, Hu A Q, Liu Q L, He X Y, Guo X. Interface-Induced High Responsivity in Hybrid Graphene/GaAs Photodetector. *Advanced Optical Materials*, 2020, 8(8): 1901741

[R4]. Yang F, Cong H, Yu K, Zhou L, Wang N, Liu Z, Li C, Wang Q, Cheng B. Ultrathin Broadband Germanium–Graphene Hybrid Photodetector with High Performance. *ACS Applied Materials & Interfaces*, 2017, 9(15): 13422-13429

[R5]. Liu C-H, Chang Y-C, Norris T B, Zhong Z. Graphene photodetectors with ultra-broadband and high responsivity at room temperature. *Nature Nanotechnology*, 2014, 9(4): 273-278

Last but not least, for the linear response in the reference Casalino, et al. *ACS Photonics*, 2018, 5(11): 4577-4585, we can see that the power range in the characterization is quite small, i.e. 0 ~ 600 nW, as shown in Figure R2. And if we limit the power range in our characterization being 100 ~ 450 nW, we can also get a linear response as depicted in Figure R2. Therefore, the linear response for the devices in *ACS Photonics*, 2018, 5(11): 4577-4585 is also probably only under the low optical power range of 0 ~ 600 nW. But we can predict that the response curve will also become flat under certain larger optical power like other state-of-art graphene-silicon photodetectors.

Figure R2. Photocurrent versus optical power of the device in *ACS Photonics*, 2018 (Figure 5b). And photocurrent versus optical power at different bias voltages of our device.

Modifications in the manuscript (text in red: modified content, text in black: original content):

In the Results - Device performances:

‘...The responsivity for the microwatt-level optical signal can exceed 1 A/W, because the trap states of the graphene-silicon interface prolonged the lifetime of the photo-induced carriers before recombination, leading to the gain which largely improved the responsivity. When optical power increases, the excited electrons contribute to fill the unoccupied states in the graphene to a certain level limited by the photon energy (wavelength). After that, extra incident power (a greater number of photons) will not be absorbed and consequently the photocurrent-power curve become flattened, together with a decreasing responsivity. ...’

3. In the Raman analysis it is not clear how they determined the doping is p instead of n [at such low doping, the 2D peak position cannot really discriminate p from n unless it is corrected for the effect of stress]. It is not clear how stress was determined. One would first need to get the doping. Then calculate what Pos(G) would correspond to that doping level in the absence of stress. Then subtract the experimental Pos(G) from that expected with no stress. This difference would provide the stress and sign. This can then be used to correct Pos(2D) to remove the effect of stress. Finally, the resulting stress-free Pos(2D) can then be used with S2 to derive the sign of the doping.

Response:

We thank the reviewer for this valuable suggestion, the peaks are indeed sensitive to the doping and stress and we make modified analysis of our experimental data according to the reviewer’s suggestion. What’s more, in our work, the doping type are determined by both the Hall effect measurement and Raman analysis. In the Hall effect measurement, the sign of the concentration indicates a p-type doping as shown in Table S1 in the SI, which also coincides with the device performances. Besides, according to the review article Nano Research, 2021, 14: 3756–3772, the graphene by PMMA assisted wet-transfer method typically results in p-type doping. Therefore, the doping type can be verified at the first place.

Figure R3. The influence of hole and electron doping on the 2D and G peaks’ parameters studied in the reference Nature Nano, 2008, 3(4): 210-215.

As for the Raman analysis, we firstly obtained the graphene doping level according to references Nature Nano, 2008, 3(4): 210-215, using the peak positions of 2D and G (pos(2D) and pos(G)), FWHM of G and intensity ratio of 2D versus G ($I(2D)/I(G)$). All these data demonstrated p-type doping level of 10^{12} cm^{-2} . However, as the reviewer suggested, these methods may not be strict ways to determine the doping level with strain effect.

Figure R4. The approach to get the strain and doping of our device using G-2D peak position map.

Therefore, we further analyzed the strain and strain-induced doping of the graphene on the device according to reference Nature Communications, 2012, 3: 1024. The key relies on the correlation between the G and 2D positions. Any given point (pos(G), pos(2D)) in the G-2D peak position map can be decomposed into two vectors along the ‘strain-free’ unit vector and ‘doping-free’ unit vector, respectively. Using the relation between strain/doping and the peak positions, the strain and doping concentration values can be obtained. The approach is summed up in Figure R4:

1. Plot the pos(G) and pos(2D) of the strain/doping-free graphene ($1,581.6, 2,676.9 \text{ cm}^{-1}$), sample points 1-7 and device point in the G-2D peak position map
2. Get the slope of the tensile strain vector by fitting the sample points 1-7, which is $\Delta\text{pos}(2D)/\Delta\text{pos}(G) = 1.97$. And plot the base line across the original point, so that we can get the tensile vector along the doping-free level. And we also plot the device line for further value reading.
3. Get the p-doping vector with slope $\Delta\text{pos}(2D)/\Delta\text{pos}(G)$ being 0.7 through the original point.
4. Determine the value ranging of the strain and doping level using the Strain-sensitivity of the G peak ($\Delta\text{pos}(G)/\Delta\epsilon = -23.5 \text{ cm}^{-1}/\%$) and Doping-level-sensitivity of the 2D peak ($\Delta\text{pos}(2D)/\Delta n =$

1.04 $\text{cm}^{-1}/10^{12}\text{cm}^{-2}$) respectively.

5. Get the strain and doping values of the device point by plotting the mapping lines.

Finally, we also get a p-type doping with concentration level of 10^{12} cm^{-2} and compressive strain of -0.06%.

Modifications in the Supplementary Information (text in red: modified content, text in black: original content):

In Section I in the Supplementary Information:

‘.....We used the van der Pauw method to perform Hall effect measurement under magnetic field intensity of 0.535 T. The electric properties of the graphene on silicon dioxide substrate are listed in Table S1. Low resistivity and high mobility were obtained. **The sign of the sheet concentration indicates a p-type doping of our sample.**

Fig. S2 Peak position of G and 2D. The two group of dashed lines represent the influences from strain (green) and doping (black, blue and red) to graphene. The blue squares are data points of points 1 to 7 with a linear fitted blue dashed line and the red squares are from the device with linear fitted red dashed line.

From the fitted Lorentzian demonstrated in Fig. S1 (b) and (c), the peak positions (pos(2D) and pos(G)), FWHM, and intensity ratio of 2D versus G ($I(2D)/I(G)$) can be obtained and calculated for doping level calculation according to the dependence curves demonstrated in reference [S2]. Using the relation between FWHM of G peak (11.42 cm^{-1}) and doping level, a concentration of about $1.7 \times 10^{12} \text{ cm}^{-2}$ was obtained. And we can verify the doping concentration using three other fitting parameters. From the pos(2D), which is 2687.66 cm^{-1} here, we can tell that the graphene integrated in the device was doped with a concentration of about $5.5 \times 10^{12} \text{ cm}^{-2}$. The results from the relation between concentration and pos(G) (1594.56 cm^{-1}) or the $I(2D)/I(G)$ (1.51) coincide well with the one obtained from the 2D peak information.

Furthermore, we analyzed the strain and strain-induced doping of the graphene on the device according to reference [S3]. The relationship between the peak position of G and 2D are demonstrated in Fig. S2. Firstly, the blue dashed line was obtained by fitting the sample points 1 to 7 with a slope of $\Delta\text{pos}(G)/\text{pos}(2D) = 1.97$ [S3]. And original point O (green solid circle, representing

strain-less and doping-neutral condition) is directly adopted from the reference [S3], being $(1581.6 \pm 0.2, 2676.9 \pm 0.7 \text{ cm}^{-1})$. The slope and original point can determine the tensile strain vector (green arrows). Secondly, direction of the doping vector can also be adopted from [S3], with a slope $\Delta\text{pos}(2D)/\Delta\text{pos}(G)$ of 0.7. Afterwards, using the Strain-sensitivity of the G peak ($\Delta\text{pos}(G)/\Delta\varepsilon = -23.5 \text{ cm}^{-1}/\%$) and Doping-level-sensitivity of the 2D peak ($\Delta\text{pos}(2D)/\Delta n = 1.04 \text{ cm}^{-1}/10^{12}\text{cm}^{-2}$) respectively, the corresponding value ranging can be determined. Finally, the strain and doping values of the device can be obtained by plotting the mapping lines, indicating a p-type doping concentration of nearly $8 \times 10^{12} \text{ cm}^{-2}$. And the strain is about -0.06%, demonstrating compressive stress which results from the photoresist cladding.'

Reviewer 2:

Authors have carefully addressed all my concerns. I believe this work will attract abroad interest of the reader of Nature Communications, I thus in favour of its publication.

Response:

We are glad that the reviewer is “in favour of its publication”. We gratefully appreciate the suggestions and comments the reviewer provided along the way, which has significantly improved the paper.

Reviewer 3:

The authors addressed all my comments efficiency and it is my belief that they enhanced the quality of the manuscript. Therefore, it is my opinion that their manuscript is fit for publication.

Response:

We are thankful for the reviewer’s opinion that “their manuscript is fit for publication”. We appreciate all the advices and comments given by the reviewer, which has substantially enhanced the quality of our paper.

REVIEWERS' COMMENTS

Reviewer #1 (Remarks to the Author):

The Authors have addressed most of the points raised by the referees. Some issues are not fully clarified, but, at this point, I would be Ok with publication.

Round 3

Reviewer 1:

The Authors have addressed most of the points raised by the referees. Some issues are not fully clarified, but, at this point, I would be Ok with publication.

Response:

We appreciate it very much that the reviewer recommended our paper for publication. We are thankful for all the advice and comments given by the reviewer, which has helped us improve the quality of our work.